# The autophagy protein ATG9A enables lipid mobilization from lipid droplets

Elodie Mailler[1], Carlos M. Guardia [1], Xiaofei Bai[2], Michal Jarnik [1], Chad D. Williamson[1], Yan Li [3], Nunziata Maio [4], Andy Golden[2] & Juan S. Bonifacino [1✉]

The multispanning membrane protein ATG9A is a scramblase that flips phospholipids between the two membrane leaflets, thus contributing to the expansion of the phagophore membrane in the early stages of autophagy. Herein, we show that depletion of ATG9A does not only inhibit autophagy but also increases the size and/or number of lipid droplets in human cell lines and *C. elegans*. Moreover, ATG9A depletion blocks transfer of fatty acids from lipid droplets to mitochondria and, consequently, utilization of fatty acids in mitochondrial respiration. ATG9A localizes to vesicular-tubular clusters (VTCs) that are tightly associated with an ER subdomain enriched in another multispanning membrane scramblase, TMEM41B, and also in close proximity to phagophores, lipid droplets and mitochondria. These findings indicate that ATG9A plays a critical role in lipid mobilization from lipid droplets to autophagosomes and mitochondria, highlighting the importance of ATG9A in both autophagic and non-autophagic processes.

[1] Neurosciences and Cellular and Structural Biology Division, Eunice Kennedy Shriver National Institute of Child Health and Human Development, National Institutes of Health, Bethesda, MD, USA. [2] Laboratory of Biochemistry and Genetics, National Institute of Diabetes and Digestive and Kidney Diseases, National Institutes of Health, Bethesda, MD, USA. [3] Proteomics Core Facility, National Institute of Neurological Disorders and Stroke, National Institutes of Health, Bethesda, MD, USA. [4] Metals Biology and Molecular Medicine Group, Eunice Kennedy Shriver National Institute of Child Health and Human Development, Bethesda, MD, USA. ✉email: juan.bonifacino@nih.gov

Autophagy is a catabolic process in which cytoplasmic materials become engulfed into double-membraned autophagosomes that eventually fuse with lysosomes for degradation by lysosomal acid hydrolases[1–3]. The process starts with de novo formation of a cup-shaped pre-autophagosomal structure termed "phagophore" or "isolation membrane" in close association with a phosphatidylinositol 3-phosphate-enriched subdomain of the endoplasmic reticulum (ER) termed "omegasome"[4–6] (see Table 1 for complete list of abbreviations). This association involves membrane continuities or membrane contact sites (MCS) between the ER and the phagophore that enable transfer of phospholipids to the phagophore membrane[6,7]. Other ER subdomains such as ER-exit sites (ERES) or organelles such as the ER-Golgi intermediate compartment (ERGIC), Golgi complex, lipid droplets (LDs), mitochondria, endosomes, lysosomes and plasma membrane are often found in the proximity of nascent phagophores and may also contribute to phagophore growth and maturation[7–14]. The phagophore membrane undergoes a rapid expansion[2,6], while sequestering cytoplasmic materials via Atg8-family proteins and cargo receptors[15], until its closure as a mature autophagosome[16,17].

The morphological and functional dissection of the early steps of autophagy has proven difficult because the phagophore membrane is rich in phospholipids but poor in integral membrane proteins that could serve as markers for the process[18,19]. Phospholipids are mainly synthesized on the cytosolic leaflet of the ER[20], consistent with this organelle being a major source for expansion of the phagophore membrane. Local synthesis of phospholipids at the ER-phagophore interface, in particular, seems to be critically important for this process[21,22]. The fatty acid (FA) precursors of phospholipids are synthesized in the cytosol and ER[23]. FAs derived from hydrolysis of triglycerides stored in LDs have also been shown to be important for synthesis of phospholipids used in phagophore membrane expansion[24–26].

The identity of the proteins involved in phospholipid delivery to the phagophore membrane has only recently begun to emerge. VMP1 and TMEM41B are two ER-associated transmembrane scramblases that mediate trans-bilayer phospholipid transport for subsequent delivery to the phagophore membrane[27–33]. In addition, the cytosolic protein ATG2 (which in humans exists as two paralogs named ATG2A and ATG2B) functions as a conduit for the transport of phospholipids between the ER and phagophore membranes[34–36]. In support of the role of all of these proteins in lipid transport in vivo, silencing of VMP1[27,29,30], TMEM41B[27,37,38] or ATG2A/B[39–41] not only prevents phagophore expansion, but also results in enlargement of LDs.

Downstream of VMP1, TMEM41B and ATG2A/B is the glycosylated, multispanning membrane protein ATG9 (Fig. 1a), which exists as a single protein in yeast (Atg9)[42] and C. elegans (ATG-9)[43], and as two paralogs named ATG9A and ATG9B in humans[44]. Whereas ATG9A is expressed in all tissues, ATG9B expression is limited to the placenta and neuroendocrine cells[44]. Unlike the ER-associated VMP1 and TMEM41B, and the cytosolic ATG2, ATG9 localizes to various post-Golgi compartments, including the trans-Golgi network (TGN), early, late and recycling endosomes, and clusters of small vesicles and thin tubules referred to as "Atg9 reservoirs" in yeast and ATG9 "vesicular-tubular clusters" (VTCs) in mammals[42,45–53]. The broad intracellular distribution of ATG9 reflects its traffic among these compartments, often in a manner regulated by autophagic stimuli[47,49,50]. ATG9 reservoirs/VTCs are located adjacent to sites of phagophore formation in the vicinity of the ER, and are believed to be the compartment whence ATG9 promotes phagophore nucleation and expansion[47,49]. At present, there is no consensus as to whether ATG9 performs these functions while remaining at ATG9 reservoirs/VTCs or upon transfer to the membrane rims of the expanding phagophore[49,54–57].

Recent cryo-electron microscopy (EM) analyses of plant[58], yeast[59] and human[60,61] ATG9 orthologs revealed that this protein is a domain-swapped homotrimer in which each protomer contributes four transmembrane α-helices and two α-helices that partially penetrate the membrane (Fig. 1a). In addition, these analyses showed that ATG9 comprises a network of internal cavities proposed to mediate transbilayer phospholipid transport[59–61]. Indeed, in vitro biochemical assays demonstrated that both yeast Atg9[59] and human ATG9A[31,61] also function as scramblases. Furthermore, structure-based molecular dynamics simulations predicted that ATG9A concentrates at highly curved membranes[60], consistent with the localization of ATG9 to small vesicles[54], ATG9 reservoirs/VTCs[47,49] and/or phagophore rims[55,56]. Finally, the structures revealed the existence of an α-helical platform on the cytosolic side of the membrane[59–61] that mediates interaction with ATG2A[60] and possibly other autophagy proteins. Despite this progress in the understanding of the structure and function of ATG9 in vitro, to date there is no evidence that ATG9 participates in any lipid transport process in vivo.

In this study, we investigated if human ATG9A has a role in intracellular lipid homeostasis, particularly in connection with LDs. Our results show that knock out (KO) or knock down (KD) of ATG9A in human cells results in increased number and size of LDs, as well as impaired transport of FAs from LDs to mitochondria and reduced mitochondrial FA β-oxidation. KO of the orthologous atg-9 in C. elegans also causes an increase in the size of LDs in hypodermal cells. In contrast, KO of ATG7, another protein that functions at a later stage of autophagy, does not affect LD abundance and mitochondrial FA transport and β-oxidation, indicating that the LD and FA metabolism defects in ATG9A-KO cells are not due to a block in autophagic degradation of LDs (i.e., lipophagy). We also demonstrate that ATG9A VTCs are closely associated with TMEM41B-enriched subdomains of the ER, and

## Table 1 Abbreviations.

| CM | Complete medium |
| --- | --- |
| CLEM | Correlative light electron microscopy |
| Co-IP | Co-immunoprecipitation |
| CRAPome | Contaminant repository for affinity purification-mass spectrometry data |
| CW | Clockwise |
| CCW | Counterclockwise |
| DIC | Differential interference contrast |
| ER | Endoplasmic reticulum |
| FA | Fatty acid |
| FACS | Fluorescence-activated cell sorting |
| FL | Full-length |
| FTS | FLAG-Two-Strep |
| gMFI | Geometric mean fluorescence intensity |
| IB | Immunoblotting |
| kDa | Kilodalton |
| KO | Knock out |
| Mr | Molecular mass |
| ns | Not significant |
| LD | Lipid droplet |
| OA | Oleic acid |
| OCR | Oxygen consumption rate |
| PD | Pulldown |
| PSC | Pearson–Spearman's correlation coefficient |
| SM | Starvation medium |
| TAG | Triacylglyceride |
| TGN | Trans-Golgi network |
| VTC | Vesicular-tubular cluster |
| WT | Wild type |

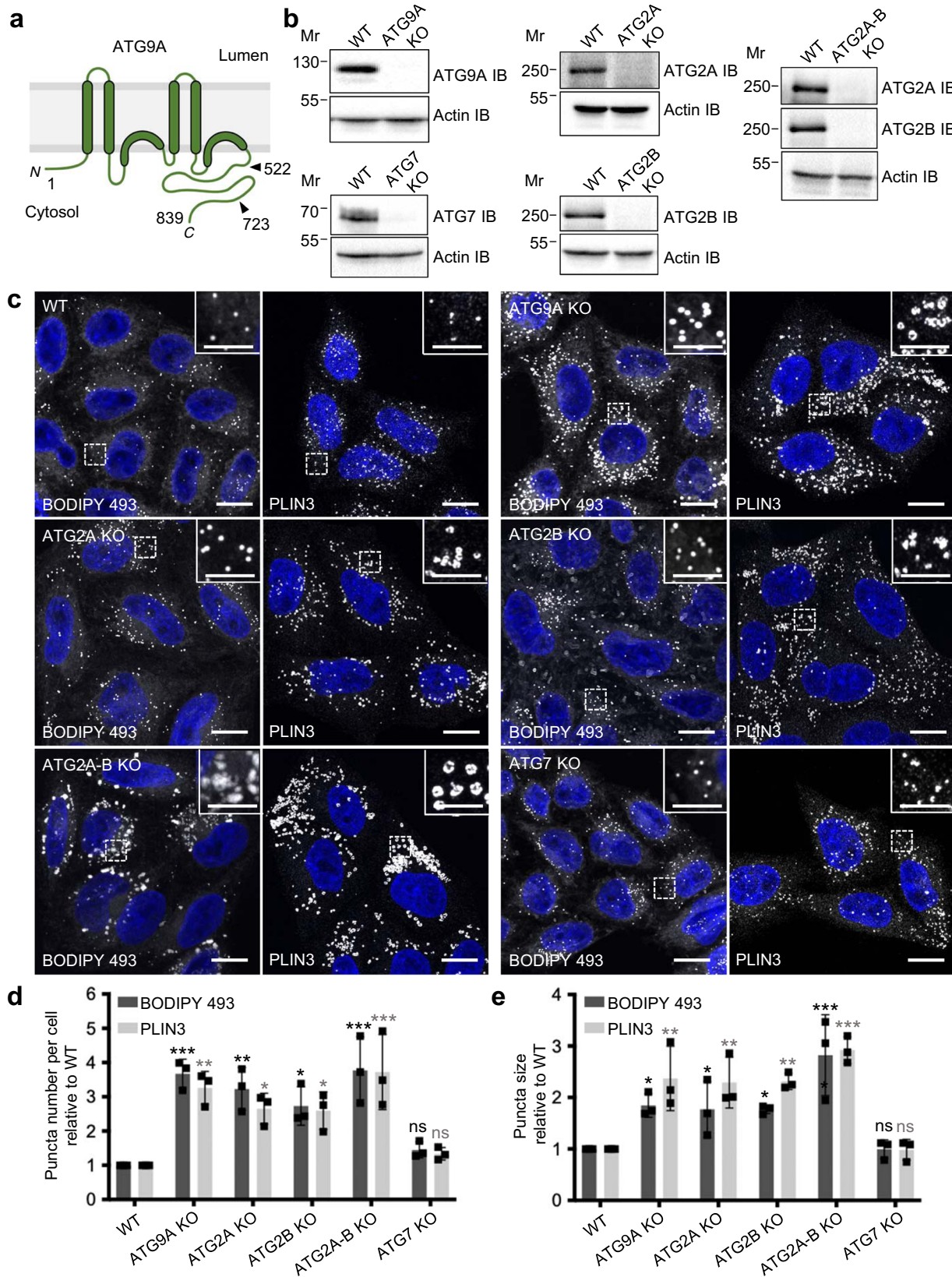

that both of these structures are found in the vicinity of phagophores, LDs and mitochondria. These findings demonstrate a requirement of ATG9A for intracellular lipid transport in vivo, and suggest that phagophore-assembly sites contribute to lipid distribution from LDs to not only expanding phagophores but also mitochondria.

## Results

**Depletion of ATG9A increases the number and size of LDs in human cells**. This project started when, in the course of our studies on ATG9A trafficking, structure and function[52,60], we noticed that CRISPR-Cas9 KO of ATG9A in the human cervical carcinoma HeLa cell line (Fig. 1b) increased the number and size

**Fig. 1 ATG9A KO increases the number and size of LDs in HeLa cells. a** Schematic representation of human ATG9A showing the four transmembrane helices, two helices that partially penetrate the membrane, and the cytosolic N- and C-terminal domains[60,61]. Numbers indicate the position of amino-acid residues relevant to this study. **b** SDS-PAGE and immunoblot (IB) analysis of WT and KO HeLa cells using antibodies to the proteins indicated on the right of each panel. Actin IB was used as a loading control. The positions of molecular mass (Mr) markers (in kDa) are indicated on the left of each panel. Results are representative from three independent experiments. **c** Confocal fluorescence microscopy of WT and KO cells that were fixed, permeabilized and stained for LDs with BODIPY 493 (green) or antibody to PLIN3 (red) (both shown in grayscale). Nuclei were stained with DAPI (blue). Scale bars: 10 μm. Insets show enlarged views of the boxed areas. Scale bars: 1 μm. Results are representative from three independent experiments. **d**, **e** The number per cell (**d**) and size (**e**) of LDs were quantified in 20 cells in each of three independent experiments using the 'Analyze particles' function of Image J. Bar graphs represent the mean ± SD fold-change of these values for BODIPY 493 (black bars) and PLIN3 (gray bars) in KO relative to WT cells. Statistical significance was determined using one-way ANOVA with Tukey post-hoc test (ns [not significant] $p > 0.05$, *$p < 0.05$, **$p < 0.01$, ***$p < 0.001$).

---

of LDs, as visualized by staining with the neutral lipid dye BODIPY 493/503 (henceforth referred to as BODIPY 493) or an antibody to the LD-associated protein PLIN3 (perilipin 3, also known as TIP47)[62] (Fig. 1c–e). Similar results were obtained upon single KO and, even more dramatically, double KO of ATG2A and ATG2B (Fig. 1b–e), in agreement with previous findings using siRNA-mediated KD of these ATG2 paralogs[41]. Transmission EM (TEM) of ATG9A-KO HeLa cells showed that LDs had a normal appearance despite their increased number and size (Supplementary Fig. 1a). KD of ATG9A in the human osteosarcoma U-2 OS cell line similarly increased LD number and size (Supplementary Fig. 1b), indicating that these effects were not limited to HeLa cells.

The increases of LDs in ATG9A-KO and ATG2A-B-KO cells could have been due to inhibition of either autophagic degradation of whole LDs (i.e., "lipophagy")[63] or non-autophagic export of lipids from LDs[24,25] to forming autophagosomes. To distinguish between these possibilities, we examined the status of LDs in HeLa cells with KO of ATG7 (Fig. 1b), an E1-like enzyme involved in conjugation of phosphatidylethanolamine to Atg8-family proteins downstream of the lipid-transport steps mediated by ATG2A/B and ATG9A[64]. ATG7 KO abrogated the conversion of the Atg8-family protein LC3B from the cytosolic LC3B-I to the membrane-tethered, phosphatidylethanolamine-conjugated LC3B-II form (Supplementary Fig. 1c), confirming the complete inactivation of the *ATG7* gene in the mutant cells. Like ATG9A and ATG2A/B, ATG7 is an essential, core component of the autophagy machinery, and its depletion blocks autophagy, including lipophagy[65]. However, ATG7 KO had no significant effect on LD number and size (Fig. 1c–e).

In line with the above results, fluorescence-activated cell sorting (FACS) showed higher BODIPY 493 staining of ATG9A-KO and ATG2A-B-KO, but not ATG7-KO, relative to wild-type (WT) cells (Fig. 2a–d; Supplementary Fig. 2), reflecting increased accumulation of neutral lipids in ATG9A-KO and ATG2A-B-KO cells. Increases in LD number and size in ATG9A-KO and ATG2A-B-KO cells occurred without significant changes in the levels of the LD-associated proteins PLIN3 and PLIN5 (also known as perilipin 5 or MLDP)[66] (Fig. 2e–h). These findings indicated a decrease in the concentration of these perilipins at the surface of LDs in the KO cells.

Taken together, the above experiments demonstrated that ATG9A and ATG2A/B maintain LD homeostasis independently of lipophagy.

**Rescue of ATG9A-KO cells demonstrates correlation of autophagy and LD phenotypes**. To confirm that increases in number and size of LDs in ATG9A-KO cells were specifically due to the lack of ATG9A, we tested the rescue of the LD phenotype by transient transfection with a plasmid encoding human ATG9A-GFP. The partial efficiency of the transient transfection allowed comparison of LDs in rescued and non-rescued cells in the same fields of view. We observed that expression of full-length

ATG9A-GFP restored normal LD number and size, whereas expression of GFP (negative control) had no effect (Fig. 2i–k). Although having only 38% amino acid identity to ATG9A, human ATG9B[44] tagged with GFP also rescued the LD phenotype in ATG9A-KO cells (Fig. 2i–k), indicating that ATG9B functions similarly to ATG9A not only in autophagy[44] but also in the maintenance of LD homeostasis.

Our previous structure-guided mutational analysis of ATG9A showed that deletion of the cytosolically-disposed, C-terminal amino acids 724 to 839 (see Fig. 1a) (ATG9A-GFP 1-723 construct), which are predicted to be largely disordered, did not affect the function of ATG9A in autophagy[60]. However, a larger deletion comprising amino acids 523 to 839 (see Fig. 1a) (ATG9A-GFP 1-522 construct), which removes part of the cytosolic platform domain, abrogated ATG9A activity in autophagy[60]. Consistently with these effects, we observed that expression of ATG9A-GFP 1-723 fully rescued both LD number and size in ATG9A-KO cells (Fig. 2i–k). In contrast, expression of ATG9A-GFP 1-522 did not rescue LD number (Fig. 2i, j). There was also a trend to less rescue of LD size in cells expressing ATG9A-GFP 1-522, but the differences did not rise to the level of statistical significance (Fig. 2i, k). These experiments thus demonstrated that the cytosolic platform domain of ATG9A, but not the downstream disordered region, is required for both autophagy and LD homeostasis, providing further evidence that both activities are linked.

**ATG9A KO phenocopies LD alterations caused by excess FAs or starvation**. The increase in LD number and size in ATG9A-KO cells is reminiscent of that observed upon incubation of WT HeLa cells with an excess of the monounsaturated FA oleic acid (OA) (Fig. 3a, b), which is stored as triacylglycerides (TAGs) in LDs[67]. We observed that OA treatment increased the already larger number and size of LDs in ATG9A-KO cells, albeit to levels similar to those in WT cells by 3 h of incubation (Fig. 3a–c). Removal of OA from the medium after 16 h incubation with OA (i.e., "chase") reduced BODIPY 493 staining in WT but not ATG9A-KO cells by 4 h of incubation (Supplementary Fig. 3), suggestive of a defect in lipid mobilization from LDs in ATG9A-KO cells.

Incubation of WT cells in medium devoid of amino acids and serum (starvation medium or SM) does not only activate autophagy[68], but also causes an increase in LD number and size due to diacylglycerol acyltransferase 1 (DGAT1)-dependent channeling of excess FAs released from autophagic degradation of intracellular organelles[69–71] (Fig. 3d–f). We found that incubation in SM did not significantly alter the already greater number and size of LDs in ATG9A-KO cells at all time points, also resulting in similar levels to those of WT cells by 3 h of SM incubation (Fig. 3d–f).

Taken together, these experiments demonstrated that ATG9A KO increases LD number/size similarly to the addition of excess FAs to the medium or to nutrient starvation of WT cells, and that

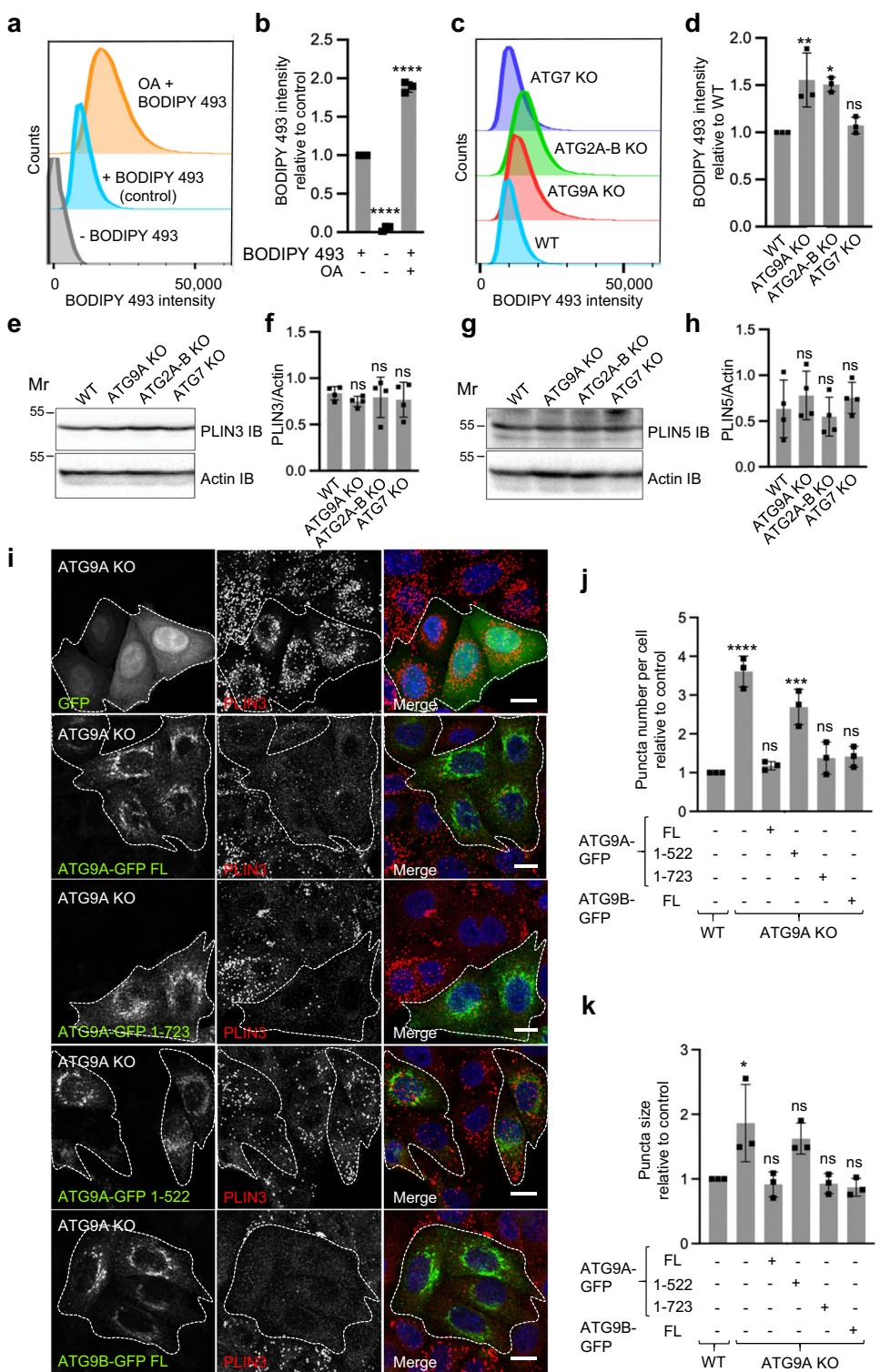

the effects of ATG9A KO and the other manipulations are largely not additive.

**ATG9A and ATG2A/B are required for mobilization of FAs from LDs to mitochondria.** The enlargement of the LD compartment in ATG9A-KO and ATG2A-B-KO cells could be due to inhibition of lipid flux from LDs to phagophores, in line with LDs being a source of lipids for expansion of the phagophore membrane[24,25] and with the roles of ATG9A[59,61] and ATG2A/B[34–36] in lipid transport in vitro. LDs also serve as reservoirs of

FAs that are transported to mitochondria and consumed by β-oxidation to produce ATP under conditions of nutrient limitation[72,73]. To investigate if ATG9A and ATG2A/B are also required for this process, we examined the mobilization of FAs from LDs to mitochondria in ATG9A-KO or ATG2A-B-KO vs. WT cells. To this end, we pulsed cells for 16 h with the fluorescent FA analog RedC12 (BODIPY 558/568 C12), chased them for 24 h in complete medium (CM) or starvation medium (SM), and examined the localization of RedC12 relative to LD (BODIPY 493) and mitochondria (TOMM20) markers (Fig. 4a)[69]. In WT cells, RedC12 remained in LDs in CM, but redistributed to

**Fig. 2 Accumulation of neutral lipids and phenotypic rescue of ATG9A-KO cells. a–d** Measurement of neutral lipid levels in WT HeLa cells by BODIPY 493 staining and FACS analysis. **a** WT cells were incubated for 2 h at 37 °C with or without 200 µM oleic acid (OA), and stained (+) or not (-) with BODIPY 493. **b** Quantification of the BODIPY 493 geometric mean fluorescence intensity (gMFI) of 100,000 cells per biological replicate in three independent experiments such as that shown in panel **a**. Bar graph represents the mean ± SD fold-change in each condition relative to non OA-incubated cells stained with BODIPY 493. Statistical significance was determined using one-way ANOVA with Tukey post-hoc test (****$p < 0.0001$). **c** WT, ATG9A-KO, ATG2A-B-KO and ATG7-KO cells were stained with BODIPY 493 and analyzed by FACS. **d** Quantification of the BODIPY 493 gMFI of 100,000 cells per biological replicate in three independent experiments such as that shown in **c**. Bar graph represents the mean ± SD fold-change in KO relative to WT cells. Statistics were as described in **b** (ns $p > 0.05$, *$p < 0.05$, **$p < 0.01$). **e–h** SDS-PAGE and IB analysis of WT and KO cells for PLIN3 (**e**) and PLIN5 (**g**). The positions of molecular mass (Mr) markers (in kDa) are indicated on the left of each panel. Results are representative from three independent experiments. Bar graphs represent the mean ± SD of PLIN3 (**f**) and PLIN5 (**h**) levels normalized for actin from four independent experiments such as those in **e** and **f**, respectively. Statistical significance was determined using one-way ANOVA with Tukey post-hoc test (ns $p > 0.05$). **i–k** Functional rescue of the LD phenotype of ATG9A-KO cells. **i** Confocal fluorescence microscopy of ATG9A-KO HeLa cells transiently transfected with plasmids encoding GFP (control), ATG9A-GFP FL (full length), ATG9A-GFP 1-522, ATG9A-GFP 1-723 or ATG9B-GFP FL (green), and stained with an antibody to PLIN3 (red) and DAPI (blue). Single-channel images are shown in grayscale. Transfected cells are outlined. Scale bar: 10 µm. Results are representative from three independent experiments. **j, k** The number per cell (**j**) and size (**k**) of LDs were quantified using the "Analyze particles" function of Image J in 20 cells in each of three independent experiments such as that shown in panel **i**. Bar graphs represent the mean ± SD fold-change of values relative to untransfected WT cells. Statistical significance was determined using one-sided ANOVA with Tukey post-hoc test (ns $p > 0.05$, *$p < 0.05$, ***$p < 0.001$, ****$p < 0.0001$).

mitochondria in SM (Fig. 4b, c). Addition of etomoxir, an inhibitor of carnitine palmitoyltransferase I (CPT1)-mediated import of FAs into mitochondria[74], blocked the transport of RedC12 from LDs to mitochondria (Supplementary Fig. 4a, b), indicating that RedC12 behaves as native FAs in this pathway. Importantly, KO of ATG9A or ATG2A-B blocked redistribution of RedC12 to mitochondria, whereas KO of ATG7 did not (Fig. 4d, e). From these experiments, we concluded that ATG9A and ATG2A/B are required for FA flux from LDs to mitochondria during starvation, but ATG7 is not. These findings mean that ATG9A and ATG2A/B promote LD-to-mitochondria FA transport independently of lipophagy, likely involving hydrolysis of LD TAGs by cytoplasmic lipases[69].

**Reduced mitochondrial FA β-oxidation in ATG9A-KO and ATG2A-B-KO cells**. The impaired mobilization of FAs from LDs to mitochondria in ATG9A-KO and ATG2A-B-KO cells prompted us to analyze if these cells have a reduced ability to metabolize FAs by mitochondrial β-oxidation. This was done by measuring the oxygen consumption rate (OCR) of cells cultured in substrate-limiting medium using a Seahorse analyzer (Fig. 5). We observed that ATG9A-KO and ATG2A-B-KO cells, but not ATG7-KO cells, had a lower basal OCR relative to WT cells (Fig. 5a–e), reflecting a decreased rate of respiration. To correct for oxygen consumption from non-mitochondrial processes, we measured the OCR upon complete inhibition of mitochondrial respiration by the addition of a combination of rotenone and antimycin A, which block mitochondrial complexes I and III, respectively. Under these conditions, OCR levels became similarly low in all the cell lines (Fig. 5a–d), indicating that differences in baseline OCR reflected specific changes in mitochondrial metabolism. Addition of etomoxir decreased the OCR in WT and ATG7-KO cells, but did not change the OCR in ATG9A-KO and ATG2A-B-KO cells, such that all cell lines exhibited similar OCR levels (Fig. 5a–e). These observations demonstrated that differences in basal OCR were due to impaired FA utilization, and that blockade of FA transfer to mitochondria in WT and ATG7-KO cells phenocopied the defects observed in ATG9A-KO and ATG2A-B-KO cells.

We also observed that addition of excess RedC12 as a FA source for 16 h increased the basal OCR in WT and ATG7-KO cells but not in ATG9A-KO and ATG2A-B-KO cells (Fig. 5b–e), demonstrating an inability to utilize excess FAs by depletion of ATG9A and ATG2A-B. The use of RedC12 as a source of FAs was validated by demonstrating a similar increase in basal OCR upon addition of OA as an alternative source (Supplementary

Fig. 4c). The increase in basal OCR upon loading with RedC12 was also inhibited by etomoxir in WT and ATG7-KO cells (Fig. 5a–e), but not in ATG9A- and ATG2A-B-KO cells, confirming that it was the result of increased mitochondrial FA metabolism in cells with an intact FA-transfer pathway.

The above experiments thus demonstrated that mitochondrial metabolism of FAs requires ATG9A- and ATG2A/B-dependent FA transport from LDs to mitochondria.

It is worth mentioning that the addition of excess RedC12 caused not only an increase in basal OCR, but also greater increases in maximal OCR (i.e., upon addition of FCCP) and non-mitochondrial OCR (i.e., upon addition of rotenone and antimycin A), in WT and ATG7-KO cells relative to ATG9A-KO and ATG2A-B-KO cells (Fig. 5b–e). In addition, proton leak (i.e., difference between OCR in cells treated with oligomycin vs. rotenone plus antimycin A) was lower in WT and ATG7-KO cells than in ATG9A-KO and ATG2A-B-KO cells (Fig. 5b–e). These observations suggest that ATG9A-KO and ATG2A-B-KO cause additional alterations in energy metabolism besides the inhibition of FA transport from LDs to mitochondria.

**Immunofluorescence microscopy shows apposition of ATG9A foci with LDs, mitochondria and ER**. The effects of ATG9A KO on LDs and mitochondria prompted us to examine the localization of ATG9A relative to these organelles by immunofluorescence microscopy of WT HeLa cells. Endogenous ATG9A showed the previously reported distribution to a perinuclear cloud characteristic of the TGN, and scattered puncta corresponding to endosomes and pre-autophagosomal structures[49–53] (Supplementary Figs. 5a and 6a). KO of ATG9A abolished the staining of both the perinuclear and peripheral populations, demonstrating that they reflected specific labeling (Supplementary Fig. 5b). Although ATG2A interacts both physically and functionally with ATG9A[31,60,75], transiently expressed GFP-ATG2A exhibited an overall different pattern, localizing to the rim of larger structures dispersed throughout the cytoplasm (Supplementary Figs. 5c and 6a). This localization was previously shown to correspond to the surface of LDs[40,41], a fact that we confirmed by co-staining for PLIN3 (Supplementary Fig. 5d). Despite the distinct patterns of ATG9A and GFP-ATG2A, we observed discrete foci where ATG9A structures were closely apposed to GFP-ATG2A structures (Fig. 6a, arrows). As the ATG9A–GFP-ATG2A co-localization was only partial, we quantified it by calculating the Pearson-Spearman's correlation coefficient (PSC) relative to two random co-localization controls[76] in which the ATG9A image was rotated 90°

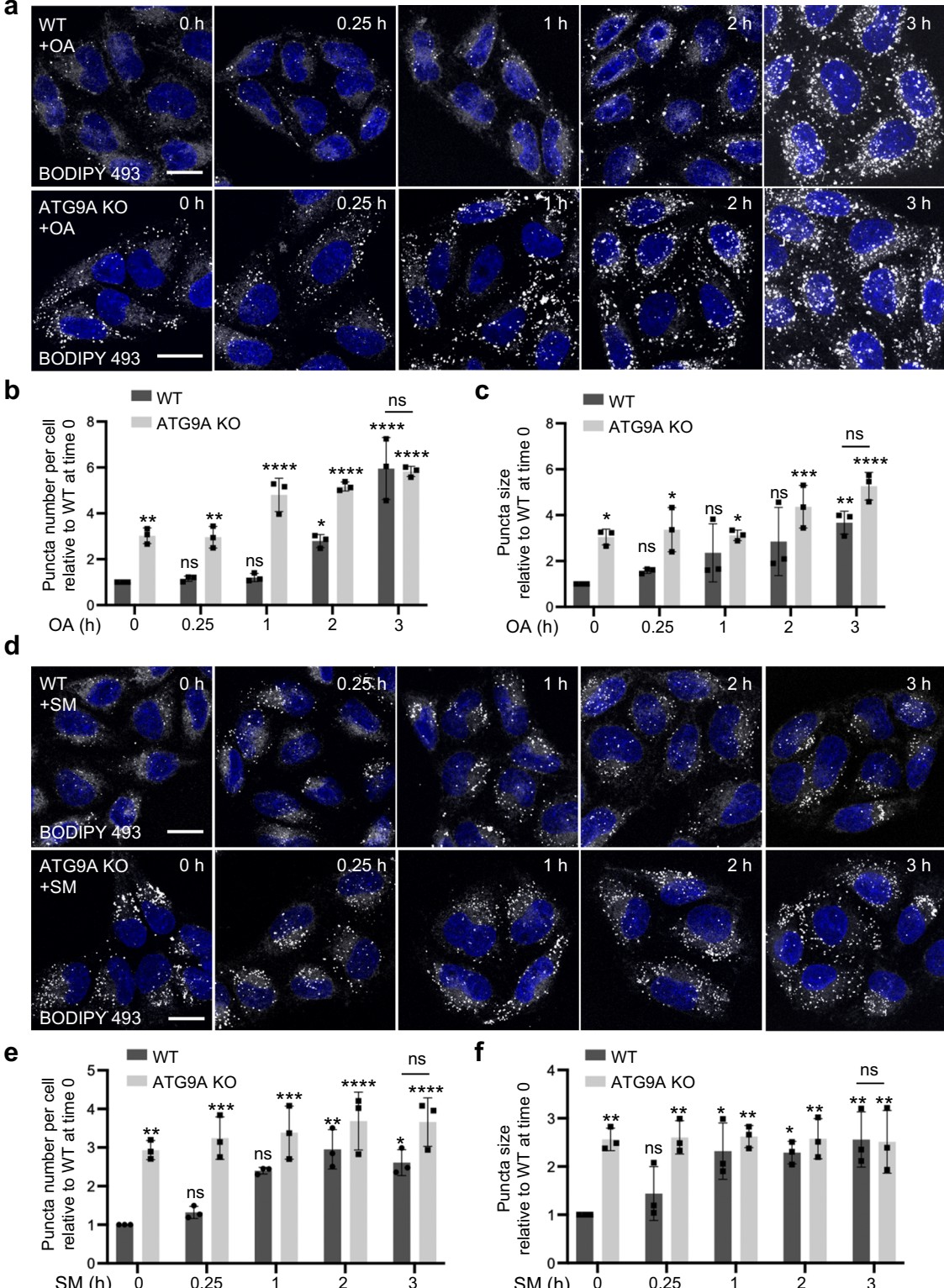

**Fig. 3 ATG9A KO phenocopies LD alterations caused by OA feeding or starvation. a** Effect of OA treatment on LDs in WT and ATG9A-KO cells. Cells were incubated for the indicated times at 37 °C with 200 μM oleic acid (OA), fixed, permeabilized, stained for LDs with BODIPY 493 (green) (shown in grayscale) and nuclei with DAPI (blue), and examined by confocal fluorescence microscopy. Scale bars: 10 μm. Results are representative from three independent experiments. **b**, **c** The number per cell (**b**) and size (**c**) of LDs were quantified in 20 cells in each of three independent experiments using the "Analyze particles" function of Image J. Bar graphs represent the mean ± SD fold-change of these values relative to WT cells at time 0. Statistical significance was determined using one-way ANOVA with Tukey post-hoc test (ns $p > 0.05$, *$p < 0.05$, **$p < 0.01$, ***$p < 0.001$, ****$p < 0.0001$). **d** Effect of starvation on LDs in WT and ATG9A-KO cells. Cells were incubated for the indicated times in amino-acid- and serum-free medium (starvation medium or SM), and analyzed by confocal fluorescence microscopy as described for panel **a**. Scale bars: 10 μm. Results are representative from three independent experiments. **e**, **f** The number per cell (**e**) and size (**f**) of LDs were quantified as described for panels **b**, **c**.

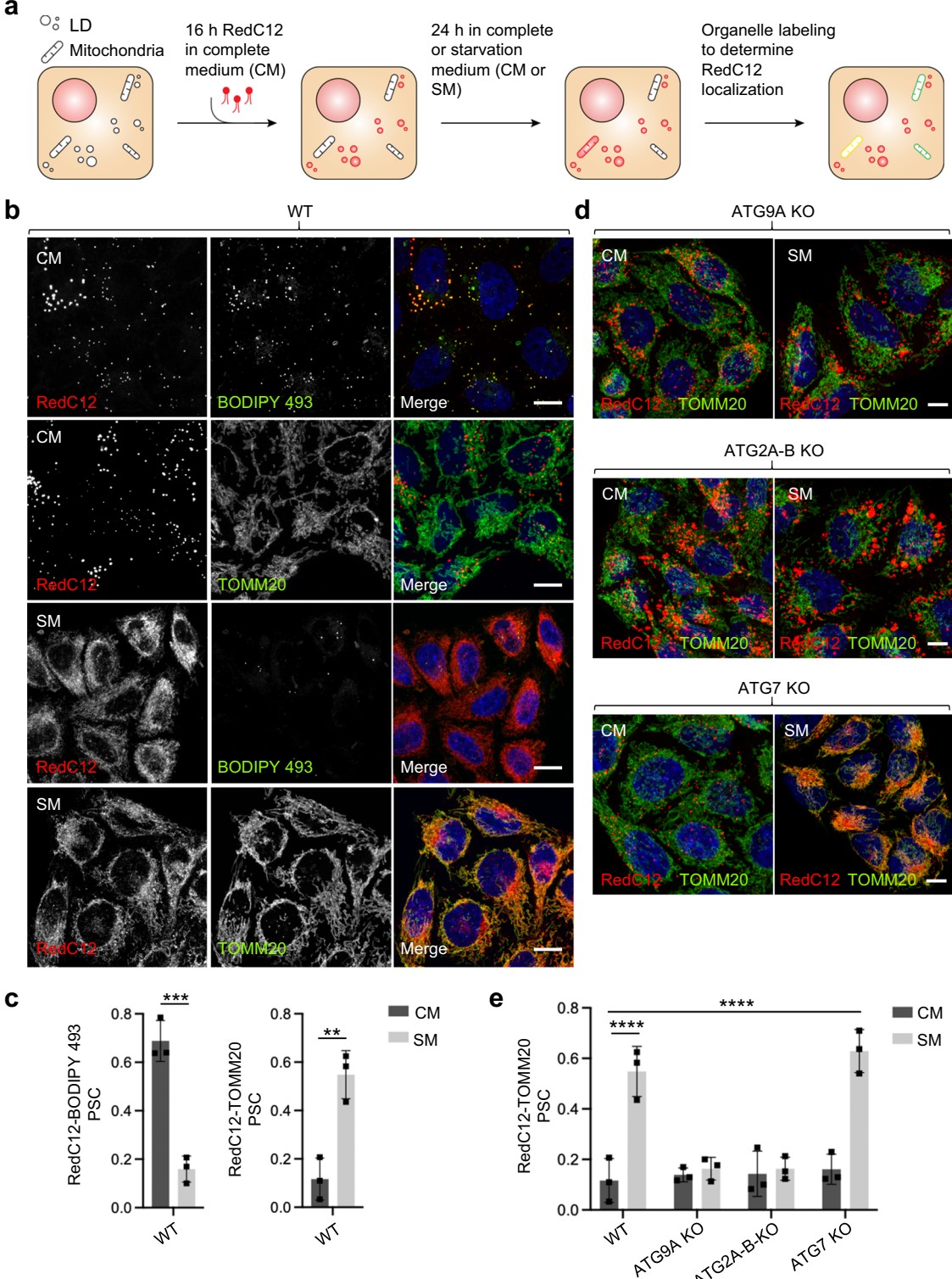

clockwise (Fig. 6a, right-most column) or 90º counter-clockwise (Supplementary Fig. 6a). This quantification yielded a PSC of approximately 0.2 for ATG9A vs. GFP-ATG2A and approximately –0.1 for both rotation controls (Fig. 6b and Supplementary Fig. 6b). Similar results were obtained for the co-localization of endogenous ATG9A with LDs labeled with BODIPY 493 (Fig. 6c,d, Supplementary Figs. 5e and 6c, d).

Further immunofluorescence microscopy analyses showed that endogenous ATG9A and mitochondria labeled with dsRed-mito

also had largely distinct distributions, although some ATG9A puncta were closely apposed to dsRed-mito-labeled mitochondria (Fig. 6e). Quantitative comparison of the patterns yielded a PSC of approximately 0.2 that decreased to approximately 0 for the 90º clockwise (Fig. 6f) and approximately –0.2 for the 90º counter-clockwise rotation controls (Supplementary Fig. 6e, f).

Finally, we compared the localization of endogenous ATG9A relative to that of the ER transmembrane protein GFP-tagged TMEM41B, which is a scramblase with dual functions in

**Fig. 4 Impairment of FA transfer from LDs to mitochondria in ATG9A-KO cells. a** Schematic representation of the microscopy pulse-chase assay for transfer of fluorescent fatty acid (FA, RedC12) from LDs to mitochondria[69]. **b** WT HeLa cells were pulsed for 16 h with RedC12, chased for 24 h in regular culture medium (complete medium or CM) or amino-acid- and serum-free medium (starvation medium or SM), fixed, stained for LDs with BODIPY 493 (red), mitochondria with antibody to TOMM20 (green), and nuclei with DAPI (blue), and imaged by confocal fluorescence microscopy. Single-channel images are shown in grayscale. Scale bars: 10 μm. Results are representative from three independent experiments. **c** Quantification of co-localization of RedC12 with LDs and mitochondria in WT cells from experiments such as that shown in panel **b**. The Pearson-Spearman's correlation coefficient (PSC) between signals in the two channels was calculated by using the PSC colocalization plugin in ImageJ. Bar graphs represent the mean ± SD from 20 cells per biological replicate in three independent experiments. Statistical significance was determined using an unpaired two-tailed Student's $t$-test (**$p < 0.01$, ***$p < 0.001$). **d** ATG9A-KO, ATG2A-B-KO, and ATG7-KO cells were analyzed as described for WT cells in panel **b**. Results are representative from three independent experiments. **e** Quantification of co-localization of RedC12 with LDs and mitochondria from experiments such as that shown in panel **d** as described for WT cells in panel **c**. Statistical significance was determined using one-sided ANOVA with Tukey post hoc test (****$p < 0.0001$).

phagophore expansion and LD metabolism[27,33,37,38]. In this case, we observed more extensive co-localization of peripheral ATG9A puncta with discrete foci containing TMEM41B-GFP (Fig. 6g), which likely correspond to an ER subdomain where TMEM41B performs its role[27,37,38]. TMEM41B-GFP additionally localized to a network devoid of ATG9A corresponding to the rest of the ER (Fig. 6g). The overall PSC of ATG9A with TMEM41B-GFP (including both punctate and reticular TMEM41B-GFP) was approximately 0.25 vs. 0 for the 90º clockwise (Fig. 6h) and –0.05 for the 90º counter-clockwise (Supplementary Fig. 6g, h) rotation controls.

Taken together, these analyses demonstrated that ATG9A-containing organelles partially but significantly co-localize with LDs, mitochondria and an ER subdomain involved in phagophore formation/expansion, the convergence of which likely facilitates the function of ATG9A in lipid flux among these organelles.

**Physical association of ATG9A- and TMEM41B-containing structures.** To further investigate the association of ATG9A and TMEM41B structures, we examined the effect of overexpressing the AP-4–interacting, kinesin-1 adaptor RUSC2, which was previously shown to redistribute ATG9A to peripheral cell protrusions (also referred to as cell vertices)[53,77]. We observed that overexpression of GFP-RUSC2, but not GFP (control), promoted peripheral redistribution of not only endogenous ATG9A but also a fraction of FLAG-Two-Strep (FTS)-tagged TMEM41B (Fig. 7a, arrows).

In addition, we performed tandem affinity purification followed by mass spectrometry (TAP-MS) of proteins that co-purify with ATG9A-FTS expressed in HeLa cells. Among the co-purifying proteins was TMEM41B, albeit with only one peptide detected in the MS analysis (Fig. 7b; Supplementary Data 1). To confirm the ATG9A-TMEM41B interaction, we tested for pulldown of endogenous ATG9A with TMEM41B-FTS or CD63-FTS (non-specific control) expressed by transfection in HeLa cells. We observed that, in WT cells, ATG9A came down with TMEM41B-FTS but not CD63-FTS (Fig. 7c, d). Importantly, double KO of ATG2A and ATG2B abrogated this pulldown (Fig. 7c, d), indicating that the ATG9A-TMEM41B interaction is dependent on ATG2A/B. This consistent with the fact that ATG2A interacts with both ATG9A[31,60,75] (Supplementary Fig. 6d, e) and TMEM41B[31]. 

From these experiments, we concluded that ATG9A-containing structures are tightly associated with TMEM41B-containing ER foci, and that this association may, at least in part, be mediated by physical interactions among ATG9A, ATG2A/B and TMEM41B.

**Correlative light-electron microscopy (CLEM) confirms the presence of ATG9A in the vicinity of LDs, mitochondria, and ER.** To gain ultrastructural insights into the location of ATG9A relative to other organelles, we performed correlative light-electron microscopy (CLEM)[78] of cells incubated in starvation medium for 20 min to induce autophagy and redistribution of ATG9A to sites of phagophore formation[47,49,50]. The methodology consisted of identifying structures containing ATG9A-mCherry and other organellar markers by fluorescence microscopy (Fig. 8a) and examining those same structures by transmission electron microcopy (TEM) of plastic-embedded serial sections (Fig. 8b–g), followed by 3D reconstruction (Fig. 8h, i). We observed that foci containing ATG9A-mCherry had the typical ultrastructure of clusters of small vesicles and tubules (i.e., VTCs) (Fig. 8b, g) located next to phagophores or autophagosomes (Fig. 8d)[49,51]. These ATG9A foci were also in close proximity to ER cisternae labeled with TMEM41B-GFP, and mitochondria labeled with mito-BFP (Fig. 8a) and recognized in TEM images by their characteristic ultrastructure (Fig. 8b–i). In addition, ATG9A foci were close to LDs, identified as dark, electron-dense globules in the TEM images (Fig. 8b–i). Additional examples of these observations, including proximity of ATG9A foci to autophagosomes, are shown in Supplementary Fig. 7. These analyses thus confirmed the confluence of these organelles at sites of autophagosome formation.

**KO of atg-9 in C. elegans causes enlargement of LDs in hypodermal cells.** Finally, to investigate if ATG9 KO also caused enlargement of LDs in a whole animal, we performed experiments with C. elegans, which expresses a single ATG-9 protein (Supplementary Fig. 8a). We observed that, in WT animals, ATG-9 endogenously tagged with GFP (ATG-9::GFP) is strongly expressed in the nerve ring (Supplementary Fig. 8b)[43], as well as in the somatic sheath and spermatheca (Supplementary Fig. 8c). We also found strong expression of ATG-9::GFP in the hypodermis (Supplementary fig. 8d), a tissue that contains abundant LDs detectable by BODIPY 493 staining (Supplementary Fig. 8e)[79]. Notably, in hypodermal cells, ATG-9::GFP puncta co-localized with structures containing the seipin ortholog SEIP-1 endogenously tagged with mScarlet (SEIP-1::mScarlet) (Fig. 9a). As seipin/SEIP-1 localizes to ER-LD contact sites[80,81], this finding is consistent with the tight association of ATG9 structures with the ER and LDs observed in cultured human cells (Figs. 6–8).

Next, we examined the effect of knocking out the atg-9 gene on LDs in hypodermal cells. To this end, we deleted all 11 exons and introns of atg-9 by CRISPR-Cas9 genome editing (Fig. 9b and Supplementary Fig. 9f–h). In all tested atg-9Δ animals, we observed a significant reduction of the number of F1 progeny when compared to WT [atg-9(+)] (Supplementary Fig. 9i). This is consistent with the strong expression of ATG-9::GFP in the somatic sheath and spermatheca, and suggesting a functional role of ATG-9 during C. elegans reproduction. Importantly, LDs in atg-9Δ hypodermal cells were enlarged relative to those in WT animals (1.07 μm vs. 0.65 μm mean diameter, respectively) (Fig. 9c, d). However, the number of LDs was not changed in atg-9Δ vs. WT hypodermal cells (Fig. 9e).

Taken together, these analyses showed that depletion of ATG9 expands the LD pool, manifested as increases in the number and size

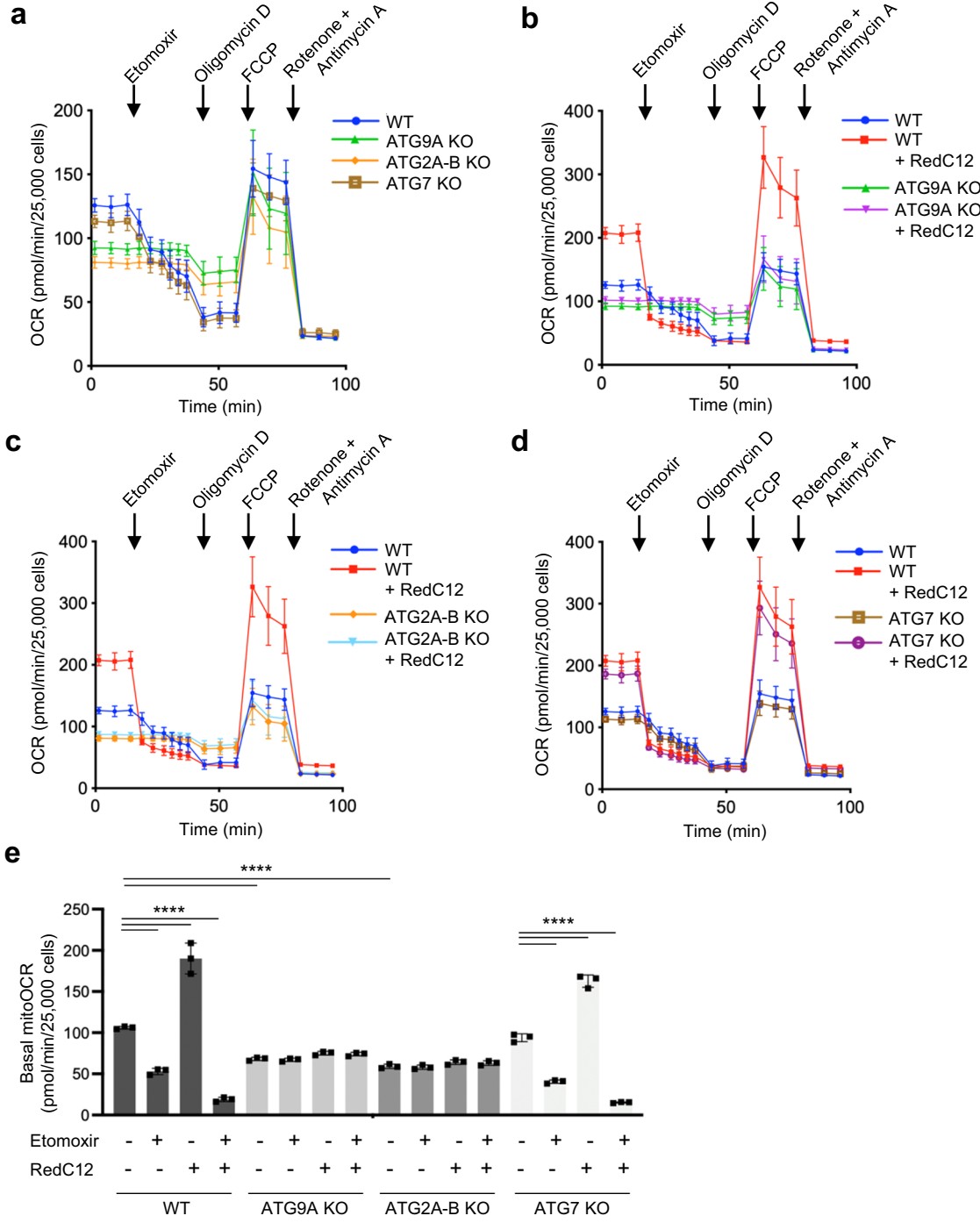

**Fig. 5 ATG9A KO and ATG2A-B KO block FA β-oxidation in mitochondria. a–e** Mitochondrial β-oxidation of FAs in WT and KO cell lines was analyzed by measuring the oxygen consumption rate (OCR) with a Seahorse flux analyzer. The contribution of FA metabolism to the OCR was determined by the addition of 4 μM of the mitochondrial FA import inhibitor etomoxir. **a** Representative Seahorse analysis of OCR in WT and the indicated KO cell lines. **b–d** Representative Seahorse analysis showing the OCR of WT and the indicated KO cell lines in the presence or absence of 1 μM RedC12 as an extra source of FAs. **e** Quantification of basal mitochondrial OCR in WT and KO cells with or without the indicated additives. Mitochondrial OCR was calculated by subtracting the OCR insensitive to rotenone plus antimycin A (last time point in **a–d**) from the basal OCR. Bar graph represents the mean ± SD of OCR values from three independent experiments. Statistical significance was determined using one-way ANOVA with Tukey post hoc test (****$p < 0.0001$).

of LDs in human cells, and of LD size in whole *C. elegans*, demonstrating the generality of the role of ATG9 in LD homeostasis.

## Discussion

Among all the components of the core autophagy machinery, ATG9 has garnered particular attention because of its unique properties. First, unlike other components of the machinery,

ATG9 undergoes vesicular transport, either directly or indirectly, between the TGN and VTCs adjacent to forming auto-phagosomes[42,45–53]. Furthermore, ATG9 functions as a scram-blase that flips phospholipids between the two monolayers of the membrane bilayer[31,59–61]. These properties endow ATG9 with the ability to participate in the delivery of phospholipids to expanding phagophores for progression of the autophagy process.

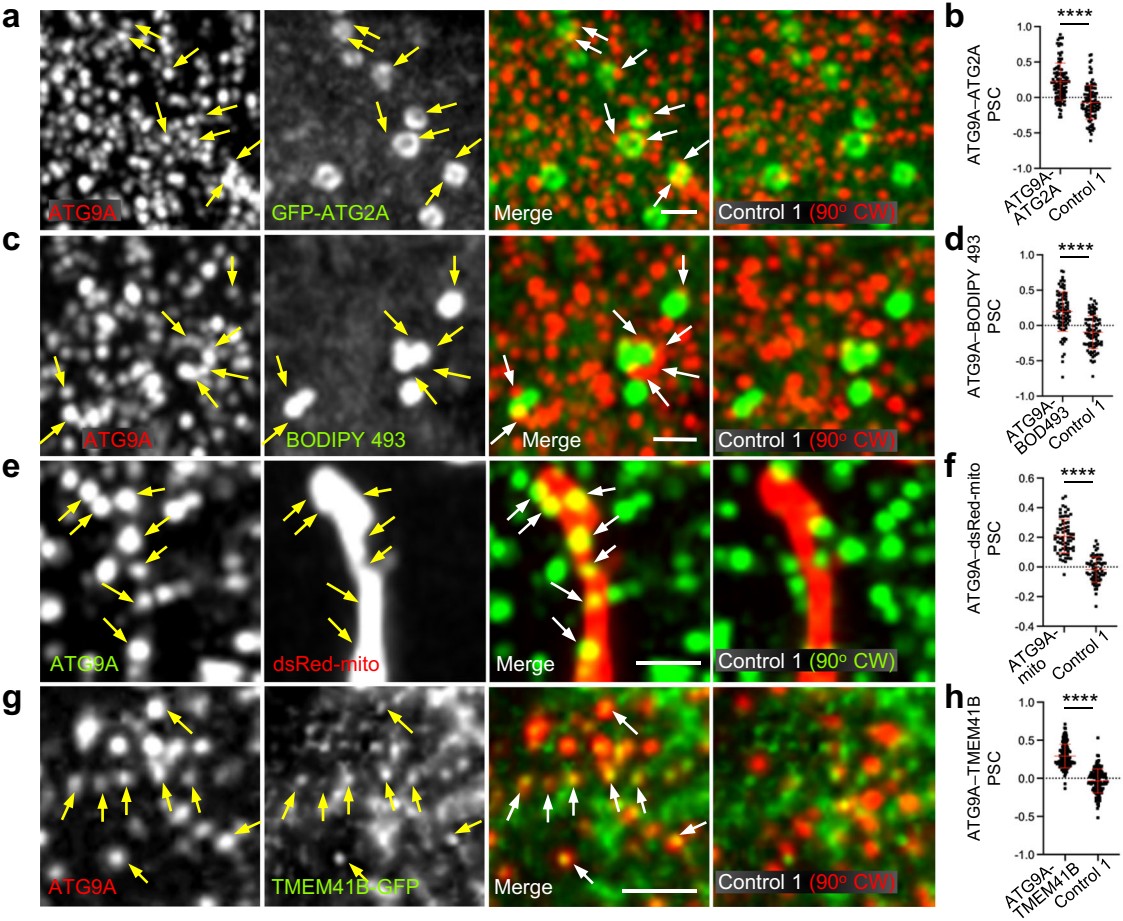

**Fig. 6 Immunofluorescence microscopy reveals apposition of ATG9A foci to LDs, mitochondria and ER. a** Confocal fluorescence microscopy of HeLa cells transiently transfected with a plasmid encoding GFP-ATG2A (green), fixed, permeabilized, and stained with an antibody to endogenous ATG9A (red). Images show single channels in grayscale, merged channels in green and red, and a random co-localization control in which the red channel was rotated 90º clockwise (CW). Images are enlargements of Box 1 in Supplementary Fig. 5c. Results are representative from three independent experiments. **b** Quantification of the co-localization of ATG9A with GFP-ATG2A by calculation of the PSC between signals in the green and red channels in experimental and control alignments. Calculations were done for several regions of interest per cell in 10 cells per biological replicate in three independent experiments ($n = 108$). Graphs show the mean ± SD of values from the three experiments. Statistical significance was determined using an unpaired two-tailed Student's $t$-test (****$p < 0.0001$). **c** Confocal fluorescence microscopy of HeLa cells stained with an antibody to endogenous ATG9A (red) and BODIPY 493 (green) as described for panel **a**. Images are enlargements of Box 1 in Supplementary Fig. 5e. **d** ATG9A-BODIPY 493 co-localization quantified as described for panel **b** ($n = 91$). **e** Confocal fluorescence microscopy of HeLa cells stained with an antibody to endogenous ATG9A (green) and dsRed-mito (red) as described for panel **a**. **f** ATG9A–dsRed-mito co-localization quantified as described for panel **b** ($n = 69$). **g** Confocal fluorescence microscopy of HeLa cells transfected with a plasmid encoding TMEM41B-GFP (green) and stained with an antibody to endogenous ATG9A (red) as described for panel **a**. **h** ATG9A–TMEM41B-GFP co-localization quantified as described for panel **b** ($n = 135$). Arrows indicate individual ATG9A foci and their corresponding positions in other images. Scale bars in all the panels: 1 µm.

In this study, we report a requirement of ATG9 for the maintenance of LD homeostasis in vivo. We show that KO or KD of ATG9A in human cell lines increases the number and size of LDs, as well as the content of neutral lipids. Furthermore, KO of the orthologous ATG-9 in *C. elegans* increases the size, though not the number, of LDs in hypodermal cells. Whether ATG9 KO increases LD number, size or both may depend on the particular characteristics of LD biogenesis and turnover in the cell types and organisms examined. In any event, these findings indicate an overall expansion of the LD storage compartment in all the systems studied here. Importantly, these phenotypes do not result from inhibition of lipophagy, since KO of a downstream, equally critical component of the core autophagy machinery, ATG7, has no effect on LD size/number. Expansion of the LD compartment may thus result from inhibition of non-destructive mobilization of lipids from LDs for utilization in phagophore expansion. These findings are in line with previous studies showing that LDs are a critical source of lipids for autophagosome biogenesis[24,25,82], and implicate ATG9 in the process. This interpretation notwithstanding, we cannot rule out that increased FA synthesis and esterification might also contribute to LD expansion in ATG9-KO cells or organisms, as recently suggested for the KO of TMEM41B in mouse liver[33].

The enlargement of the LD compartment in ATG9-depleted cells is similar to that observed upon silencing of VMP1[27,29,30], TMEM41B[27,37,38] or ATG2A/B[41] (this study), and contrasts with the normal appearance of LDs in cells silenced for ATG5[41] or ATG7[37] (this study). Recent studies demonstrated that VMP1 and TMEM41B are ER-resident phospholipid scramblases[31–33], and that ATG2A is a cytosolic lipid transport protein[34–36]. Together, these findings support the notion that VMP1/ TMEM41B, ATG2A/B and ATG9A function as a module for lipid transport from the ER to expanding phagophores, and that LDs serve as an upstream source of lipids for this process.

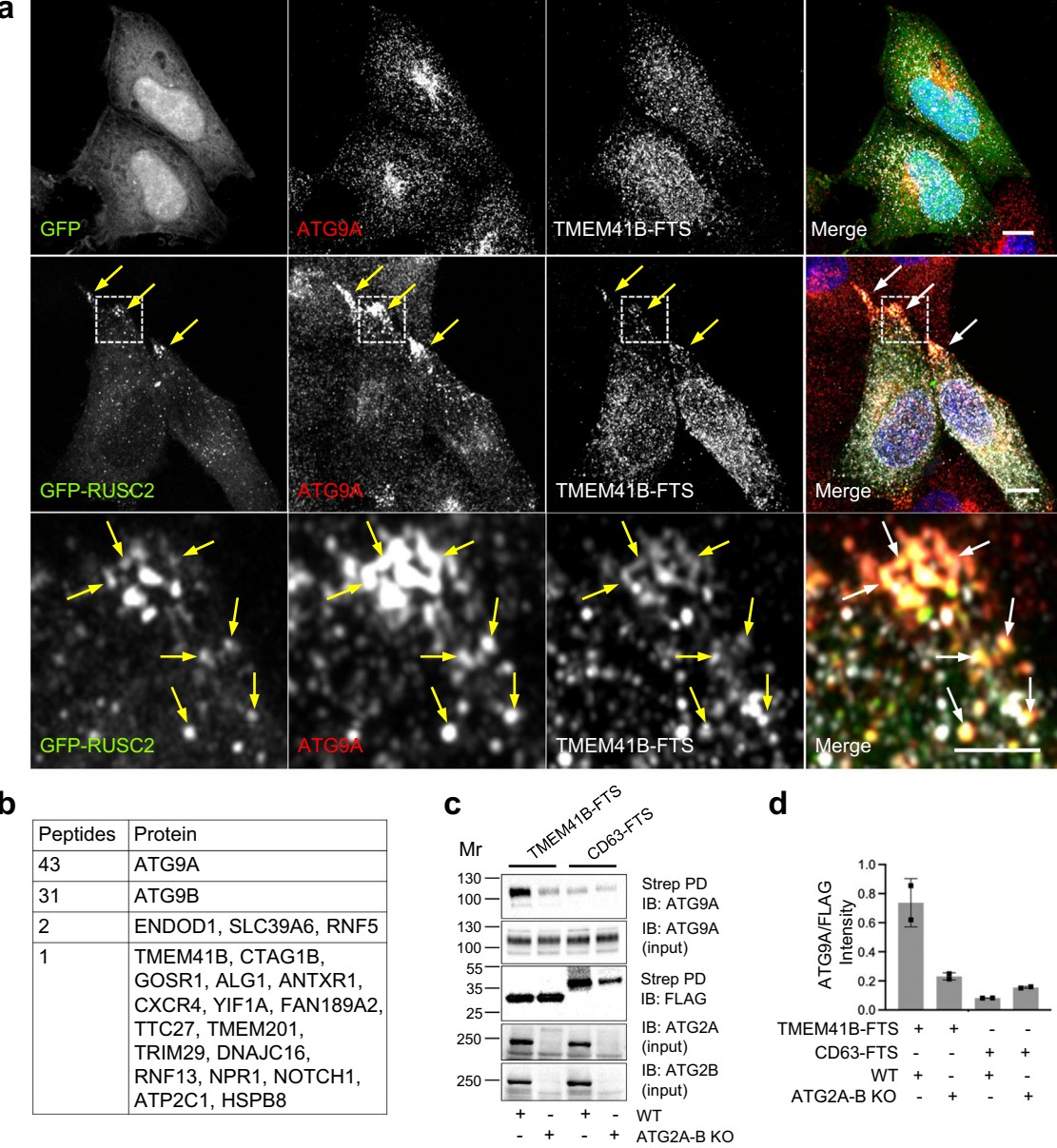

**Fig. 7 Association of ATG9A and TMEM41B structures. a** Redistribution of ATG9A and TMEM41B-FTS to cell vertices by overexpression of GFP-RUSC2. Confocal fluorescence microscopy of HeLa cells transiently transfected with plasmids encoding GFP (control) or GFP-RUSC2 (green) and TMEM41B-FTS, fixed, permeabilized and stained with antibodies to endogenous ATG9A (red) and FLAG epitope (to label TMEM41B-FTS) and with DAPI (blue). Single-channel images are shown in grayscale. Scale bars: 10 µm. Enlarged views of the boxed areas are shown in the bottom row. Scale bar: 1 µm. Co-localization at cell vertices is indicated by arrows. Results are representative from three independent experiments. **b** Summary of TAP-MS analysis of proteins that co-purify with ATG9A-FTS. Raw data are shown in Supplementary data 1. **c** Validation of ATG9A-TMEM41B interaction by Strep-Tactin pulldown (Strep PD) and IB. WT and ATG2A-B-KO HeLa cells were transiently transfected with plasmids encoding TMEM41B-FTS or CD63-FTS (non-specific control). Cell extracts were incubated with Strep-Tactin beads, and bound and input proteins analyzed by SDS-PAGE and immunoblotting with antibodies to ATG9A, ATG2A, ATG2B and the FLAG epitope. The positions of molecular mass (Mr) markers (in kDa) are indicated on the left. **d** Quantification of the ratio of ATG9A in the PD to FLAG in the input in two experiments such as that shown in **c**. Bars represent the means ± SD from two independent experiments.

Although ATG9A, ATG2A/B and TMEM41B are associated with different compartments and have different overall cytoplasmic distributions, they converge at sites of phagophore formation. The best co-localization that we observed by fluorescence microscopy was between ATG9A and TMEM41B. ATG9A exhibits a rather unique intracellular distribution, consisting of a perinuclear cloud that co-localizes with TGN markers, and a scattering of fine puncta that co-localize with TMEM41B-enriched ER foci. This latter co-localization represents not just spatial proximity, but physical association of ATG9A- and TMEM41B-containing structures, since both proteins can be jointly redistributed to the cell periphery upon overexpression of RUSC2. Moreover, ATG9A and TMEM41B co-precipitate, albeit in an ATG2A/B-dependent manner. These findings suggest that ATG9A, TMEM41B and ATG2A/B are part of a physical, supramolecular complex involved in lipid transport between the ER and VTCs. At present we do not know if interactions between these proteins are direct or indirect, and if other proteins are

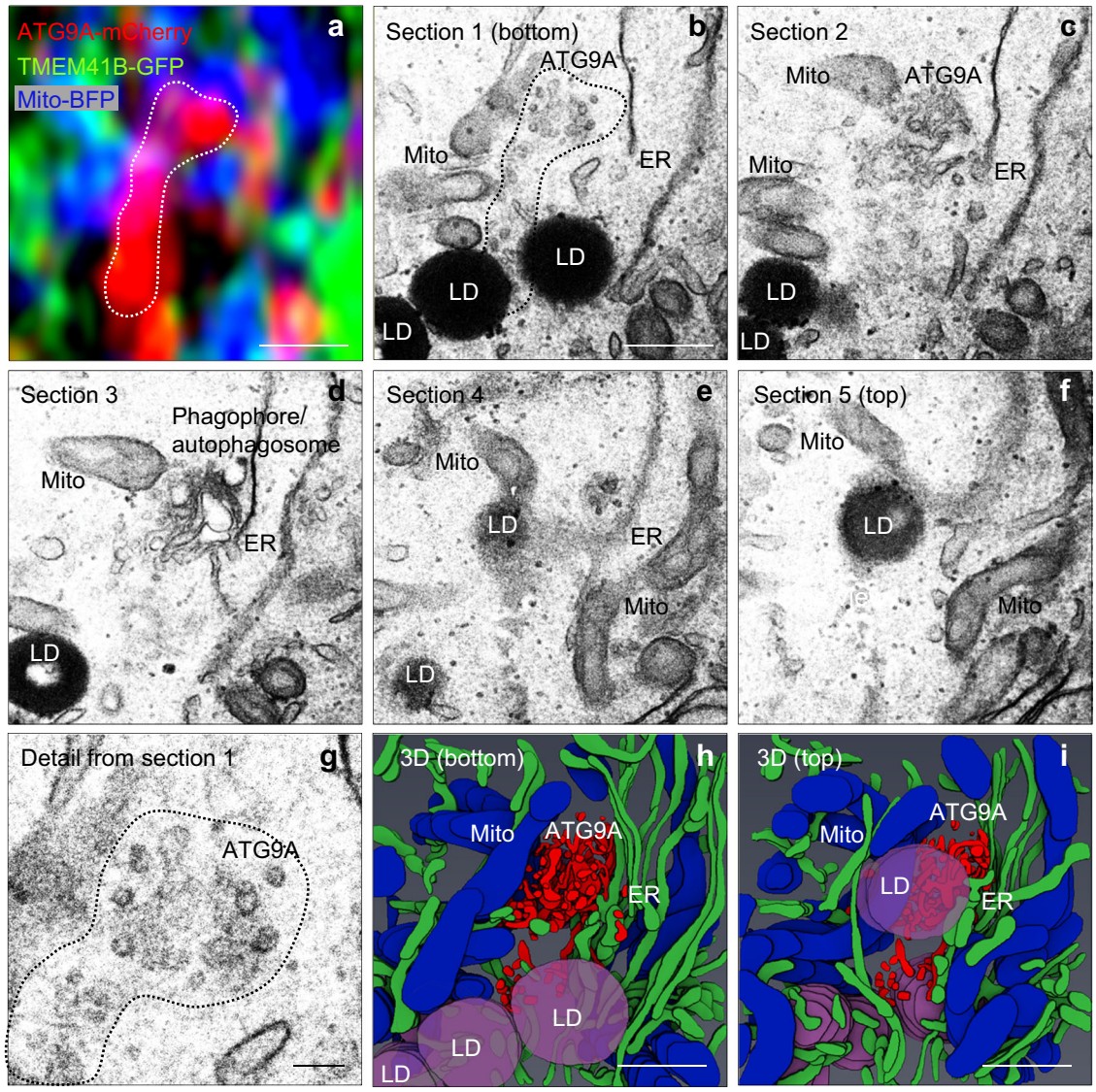

**Fig. 8 Correlative light-electron microscopy of ATG9A localization relative to other organelles.** HeLa cells transiently transfected with plasmids encoding ATG9A-mCherry (red), TMEM41B-GFP (green) and mito-BFP (blue) were starved for 20 min prior to fixation and analysis by CLEM. **a** Airyscan image of an ATG9A-positive structure (outlined) in close proximity to TMEM41B and mitochondria. Scale bar: 0.5 μm. **b** Transmission EM (TEM) image corresponding to the Airyscan image shown in panel **a**. Scale bar: 0.5 μm. **c**–**f** TEM images of sequential serial sections. **g** Enlarged view of the ATG9A-positive structure in **b** showing the typical appearance of a VTC. Scale bar: 0.1 μm. **h**, **i** 3D reconstruction of TEM images in panels **b**–**f** using Amira software, viewed from the bottom (**h**) or the top (**i**) of the cell. The identity of different organelles is indicated. Scale bar: 0.5 μm. Results are representative from three independent experiments.

involved. A molecular dissection of these interactions will require detailed biochemical and structural analyses, including work with purified recombinant proteins.

CLEM additionally showed that typical ATG9A VTCs occur in the vicinity of not only TMEM41B-containing ER elements, but also morphologically recognizable phagophores/autophagosomes, LDs and mitochondria. Also importantly, in *C. elegans* hypodermal cells, ATG-9 puncta co-localize with foci containing the seipin ortholog SEIP-1, an oligomeric integral membrane protein that facilitates lipid transport at ER-LD contacts[30,80,83]. The convergence of all of these organelles at sites of phagophore formation suggests a plausible pathway for lipid transport from LDs to forming autophagosomes. This pathway starts with release of FAs from TAGs stored in LDs by LD-associated lipases (i.e., lipolysis)[84]. The released FAs are then transferred to the ER, where they are used in phospholipid biosynthesis[20]. These phospholipids are next scrambled by TMEM41B (and

VMP1)[31,32] and channeled through ATG2A/B[34–36] to VTCs, where ATG9A scrambles them between the two membrane monolayers[59–61]. From the VTCs, scrambled phospholipids are delivered, by still unknown mechanisms, to the growing phagophore membrane. Disruption of this pathway might explain the autophagic defects observed in cells with mutation or depletion of LD-associated lipases[24,25], TMEM41B[27,37,38], ATG2A/B[39–41], or ATG9A[49,60,85].

Strikingly, TMEM41B[37], ATG2A/B (this study) and ATG9A (this study) are also important for transport of FAs from LDs to mitochondria, and for use of FAs in mitochondrial respiration. These findings are consistent with the presence of mitochondria in the vicinity of organelles housing these proteins. The mechanisms involved, however, are unclear. ATG2A/B is a cytosolic protein that, at least when overexpressed, can peripherally associate with LDs[40,41] (this study) and mitochondria[75]. However, at present there is no evidence that TMEM41B and

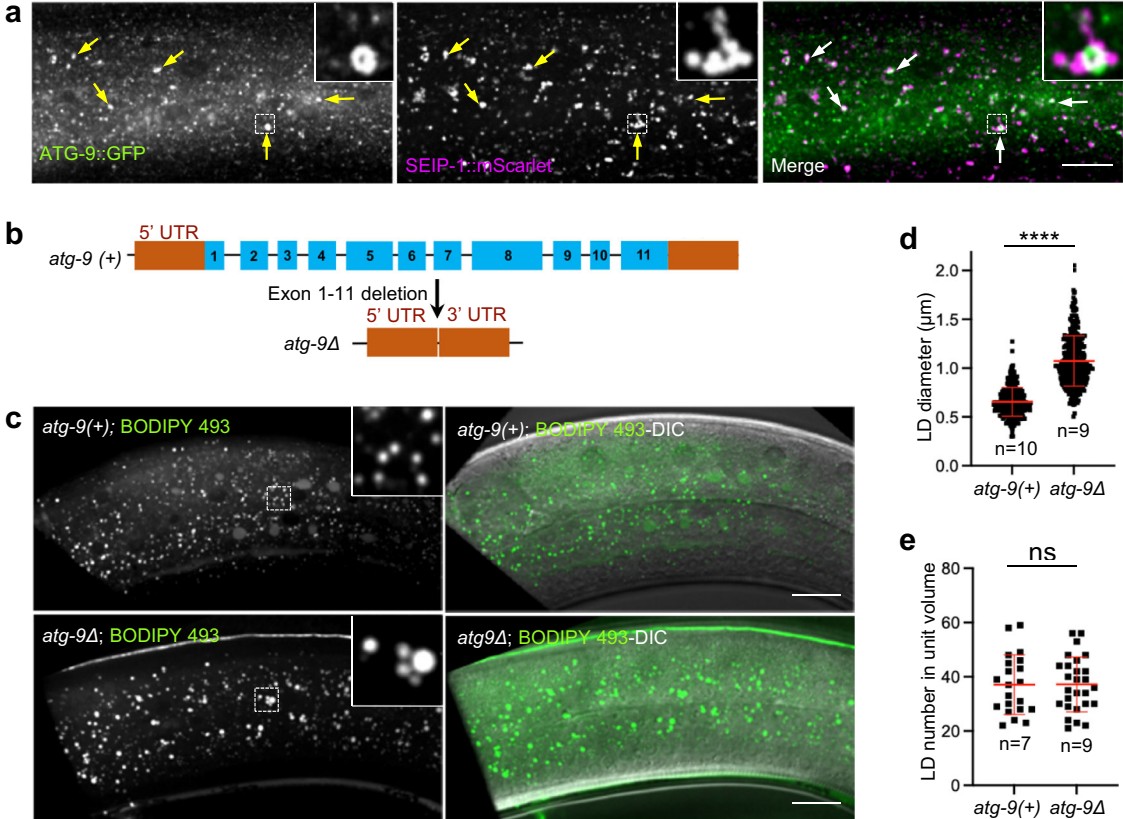

**Fig. 9 Deletion of *atg-9* causes enlargement of LDs in *C. elegans* hypodermal cells. a** Fluorescence microscopy imaging of hypodermal tissue from WT *C. elegans* expressing endogenously tagged ATG-9::GFP (green) and SEIP-1::mScarlet (magenta). Scale bar: 20 μm. Arrows indicate puncta where ATG-9::GFP and SEIP-1::mScarlet co-localize. Insets are 4x magnified views of the boxed areas. Results are representative from three independent experiments. **b** Schematic representation of the genomic structure of the WT *atg-9(+)* and mutant *atg-9Δ* alleles. **c** Fluorescence microscopy and differential interference contrast (DIC) imaging of hypodermal tissue from WT (*atg-9(+)*) and *atg-9Δ* animals stained with BODIPY 493. Scale bars: 20 μm. Insets are 4x magnified views of the boxed areas. Results are representative from three independent experiments. **d, e** Quantification of LD diameter and number in WT (*atg-9(+)*) and *atg-9Δ* hypodermal tissue in the indicated number (n) of animals. **d** Graph showing the individual values and the mean ± SD of LD diameter. Approximately 30 LDs were measured per animal. Statistical significance was determined using the unpaired two-tailed Student's *t*-test (****$p < 0.0001$). **e** Graph showing the individual values and the mean ± SD of LD number, quantified in a unit volume (*L* 18 μm x *W* 18 μm x *H* 10 μm) with three independent units measured per animal.

ATG9A localize to anywhere other than the ER and VTCs, respectively, even when these compartments are in close proximity to LDs and mitochondria. The requirement of the TMEM41B-ATG2A/B-ATG9A module could mean that flow of FAs from LDs to mitochondria involves passage through the ER and VTCs. Against this interpretation, however, is the observation that ATG2A/B has high affinity for phospholipids[36,86], but low affinity for the FA palmitic acid[35]. Likewise, TMEM41B and ATG9A are phospholipid scramblases[31–33,59,61], and are not known to flip FAs. It's also unknown at what stage of this pathway FAs would be converted to fatty acyl-CoA, which is the actual substrate for mitochondrial import of FAs by the carnitine system. These considerations make it unlikely that the role of the TMEM41B-ATG2A/B-ATG9A ensemble in LD-to-mitochondria FA transport is direct. Rather, disruption of the TMEM41B-ATG2A/B-ATG9A module could lead to an accumulation of unflipped or untransported phospholipids in the ER, indirectly altering overall lipid homeostasis in the ER, LDs and mitochondria. For example, there could be alterations in the phospholipid composition of the LD monolayer or mitochondrial membranes that ultimately impair FA transport and β-oxidation. A distinction between these alternative scenarios will require a better understanding of the function of the TMEM41B-ATG2A/

B-ATG9A module and of the mechanisms by which FAs are transported between the ER and mitochondria.

Our findings further highlight the critical importance of ATG9 for non-autophagic processes. In addition to the mitochondrial FA import and β-oxidation shown here, ATG9 has been previously implicated in the regulation of the actin cytoskeleton[87], dsDNA-induced innate immune response[85], necrotic programmed cell death[88], c-Jun N-terminal kinase (JNK)[89] and mTOR signaling[90], HIV-1 infectivity[91], and protection from plasma membrane damage[92], in most cases independently of its role in autophagy. It is tempting to speculate that ATG9 participates in these processes through its intrinsic ability to facilitate lipid distribution among various organelles.

## Methods

**Recombinant DNAs**. Complementary DNAs (cDNAs) encoding human ATG9A (WT and deletion mutants 1–723, 1–522[60]), human ATG9B and human TMEM41B were cloned into pEGFP-N1 (Clontech). Human RUCS2 cDNA was cloned into pEGFP-C1 (Clontech) and human ATG9A cDNA in pmCherry-N1 (Clontech). cDNAs encoding human ATG9A, human TMEM41B and human CD63 appended with a single FLAG-Two-Strep (FTS) tag at the C-terminus were cloned into pcDNA3.1 (Thermo Fisher Scientific). A cDNA encoding human ATG2A with an N-terminal GFP tag (GFP-ATG2A) cloned in the pMRXIP vector was a gift from Noboru Mizushima (Addgene, 114462). A pDsRed2 plasmid encoding dsRed-Mito was obtained from Clontech, and a pAcGFP1-N1 plasmid

encoding mito-BFP was a gift from Gia Voeltz (Addgene, 49151). The sequence of all constructs was confirmed by Sanger DNA sequencing.

**Antibodies and fluorescent probes**. We used primary antibodies to the following proteins (antibody sources, catalog numbers and working dilutions are indicated in parentheses): ATG2A (rabbit polyclonal, Proteintech, 23226-1-AP, 1:500), ATG2B (rabbit polyclonal, Proteintech, 25155-1-AP, 1:500), ATG7 (rabbit monoclonal, Cell Signaling, 8558, 1:1,000), ATG9A (rabbit monoclonal, Abcam, 108338, IB: 1:2,000, IF: 1:250), LC3B (rabbit monoclonal, Cell Signaling, 3868, 1:1,000), PLIN3 (mouse polyclonal, Proteintech, 10694-1-AP, 1:500), PLIN5 (guinea pig polyclonal, PROGEN, GP31, 1:500), TGN46 (sheep polyclonal, Bio-Rad, AHP500G, IF: 1:500), and TOMM20 (mouse monoclonal, Abcam, 56783, IF: 1:500). We also used the following secondary antibodies: Alexa Fluor 488-conjugated donkey anti-rabbit IgG (Invitrogen, A21206, 1:1,000), Alexa Fluor 488-conjugated donkey anti-mouse IgG (Invitrogen, A21202, 1:1,000), Alexa Fluor 555-conjugated donkey anti-rabbit IgG (Invitrogen, A31572, 1:1,000), Alexa Fluor 555-conjugated donkey anti-mouse IgG (Invitrogen, A31570, 1:1,000), Alexa Fluor 555-conjugated donkey anti-sheep IgG (Invitrogen, A21436, 1:1,000), HRP-conjugated anti-GFP (MACS, 130091833, 1:2,000), HRP-conjugated anti-actin (Sigma, A3854, 1:10,000), HRP-conjugated anti-FLAG M2 (Sigma, A8592, 1:1,000), HRP-conjugated donkey anti-rabbit IgG (GE Healthcare, NA934V, 1:5,000), HRP-conjugated sheep anti-mouse IgG (GE Healthcare, NXA931, 1:5,000), HRP-conjugated donkey anti-guinea pig IgG (Jackson Immuno Research, 706-035-148, 1:5,000), and HRP-conjugated sheep anti-mouse IgG (GE Healthcare, NXA931, 1:5,000). Fluorescent lipid probes used were: BODIPY 493/503 (referred to as BODIPY 493) (Thermo Fisher Scientific, D3922) and BODIPY 558/568 C12 (referred to as RedC12) (Thermo Fisher Scientific, D3835).

**Cell culture, transfection, infection, and treatments**. Human HeLa, U-2 OS and HEK293T cells were grown in Dulbecco's-modified Eagle's medium (DMEM) supplemented with 10% fetal bovine serum (FBS), 100 IU/ml penicillin, 100 μg/ml streptomycin, and 2 mM L-glutamine (referred to as complete medium or CM) at 37 °C and 5% $CO_2$. Transient plasmid transfection of HeLa cells was performed using Lipofectamine 2000 (Thermo Fisher Scientific, 11668019) according to the manufacturer's instructions. Transfection of siRNAs (Dharmacon) in U-2 OS was performed with Lipofectamine RNAiMAX (Thermo Fisher Scientific, LMRNA015). A second transfection of siRNA was performed 48 h after the first one, and experiments were done 24 h later.

To generate a stable cell line expressing ATG9A-FTS, retrovirus particles were prepared by transfecting HEK293T cells with pQCXIP-ATG9A-FTS and retrovirus packaging plasmids (pVSV-G, pGag-Pol) (Clontech). Supernatants containing viruses were collected 48 h after transfection and centrifuged for 10 min at 1,000 x g to remove debris. HeLa cells (300,000 per well) were incubated in a 6-well plate with 1 ml of retroviral supernatant supplemented with 1 ml CM and 5 μg polybrene, and centrifuged at 1,540 x g for 30 min, followed by incubation for 4 h at 37 °C. One well was prepared in parallel with CM without virus as a control. Supernatants were then discarded and replaced by CM. Two days later, cells were trypsinized and seeded in duplicate (300,000 per well) into a 6-well plate. One day later, one well was supplemented with 1 μg/ml puromycin, while the other well was maintained in CM. When puromycin killed all the cells in the no-virus condition, immunofluorescence and immunoblot analyses were performed to characterize the stably transduced cells.

Starvation was performed by incubating cells with Earle's balanced salt solution (EBSS, Thermo Fisher Scientific, 24010043, referred to as starvation medium or SM) for the indicated times. Cells were briefly washed once with EBSS before incubation. OA treatment was performed by incubating cells with 200 μM OA (from a stock solution dissolved in methanol) (Sigma Aldrich, O1008) for the indicated times (Fig. 3a–c), or by incubating cells for 16 h with 1 μM OA followed by three washes with CM (Supplementary Fig. 3). Inhibition of lysosomal degradation was performed by treatment with the v-ATPase inhibitor bafilomycin A1 (BafA1) (100 nM BafA1 in DMSO) for 4 h. A control condition with DMSO treatment was performed in parallel (Supplementary Fig. 1). Treatment of cells with etomoxir (4 μM in $H_2O$) was performed for 24 h in CM or SM prior to fixation and immunofluorescence. A control condition with $H_2O$ treatment was performed in parallel (Supplementary Fig. 4a, b).

**CRISPR-Cas9 HeLa cell KOs**. KO of ATG9A, ATG2A, ATG2B, ATG2A/B and ATG7 in HeLa cells was performed using the CRISPR-Cas9 system. Guide RNAs sequences (Supplementary Table 1) used to target the *ATG9A*, *ATG2A*, *ATG2B* and *ATG7* genes were cloned separately into pSpCas9(BB)-2A-GFP (pX458) (gift of Feng Zhang, Addgene, 48138). HeLa cells were transfected with plasmids containing the different targeting sequences. One day later, GFP-positive cells were selected on a FACS Aria II Flow Cytometer and single-cell clones were isolated in 96-well plates. After 14 days, clones were analyzed by immunoblotting to confirm the KO of the corresponding gene. ATG2A-KO HeLa cells were further transfected with a pX458 vector containing the guide RNA targeting the ATG2B gene to generate the ATG2A-B-KO cell line.

**SDS-PAGE and immunoblotting**. Protein concentration was measured using the Bradford protein assay (Bio-Rad). Samples were denatured in sample buffer for 5 min at 37 °C (for ATG9A) or 95 °C (for other proteins), resolved by SDS-PAGE, and transferred onto a nitrocellulose membrane (Bio-Rad). Membranes were saturated with 3% blotting-grade blocker (Bio-Rad) in PBS containing 0.1% Tween 20 (Sigma-Aldrich) for 2 h at 4 °C and incubated with the indicated primary antibodies overnight at 4 °C. HRP-conjugated secondary antibodies were incubated with the membrane at room temperature for 1 h. HRP was detected using the Clarity or Femto ECL kit (Thermo Fisher Scientific). Images were captured using a ChemiDoc system (Bio-Rad).

**Confocal fluorescence microscopy**. HeLa and U-2 OS cells were cultured on coverslips in a 24-well plate, transfected, cultured for an additional 24 h, and fixed for 15 min at room temperature with 4% paraformaldehyde (PFA) in PBS containing 0.1 mM $CaCl_2$ and 1.0 mM $MgCl_2$ buffer (PBSCM). Cells were then washed once in PBSCM and permeabilized in PBSCM containing 0.1% saponin and 1% BSA for 10 min at room temperature. Primary antibodies and Alexa-conjugated secondary antibodies were diluted in PBSCM containing 0.1% saponin and 1% BSA and sequentially incubated for 30 min at 37 °C. LD labeling was done by incubating fixed and permeabilized cells with BODIPY 493 (stock solution in DMSO, used at 2 μg/ml in PBSCM containing 0.1% saponin and 1% BSA) for 30 min at 37 °C. When combined with immunofluorescent labeling, BODIPY 493 was added to the secondary antibodies. Coverslips were mounted on glass slides with DAPI Fluoromount-G (EMS). Confocal images were obtained using the Airyscan mode of a Zeiss LSM 880 microscope with a Plan Apochromat 63x objective.

**Quantification of fluorescence microscopy images**. To quantify the number and size (area in μm²) of LDs, confocal imaging data were analyzed using the Analyze Particles plugin of Fiji/ImageJ[93]. Images were thresholded at 0.1% of total intensity to reduce the background.

The Pearson-Spearman's correlation coefficient (PSC) between signals in two channels was obtained with the Pearson-Spearman correlation (PSC) co-localization plugin in Fiji/ImageJ[94]. A default threshold value of 10 above the background fluorescence was used for both channels. The PSC indicates the extent of co-localization, with 0 and below indicating no co-localization, and 1 indicating complete co-localization.

**Transmission electron microscopy (TEM)**. HeLa cells were grown on fibronectin-coated 18-mm or, for CLEM experiments, 25-mm gridded (EMS) coverslips, and transfected as described above. For the experiment presented in Supplementary Fig. 1a, samples were fixed in 2.5% glutaraldehyde, 2% formaldehyde, 2 mM $CaCl_2$ in 0.1 M cacodylate pH 7.4 for 15 min at room temperature followed by 45 min on ice. For CLEM experiments (Fig. 8 and Supplementary Fig. 7), samples were fixed in 4% formaldehyde, 0.1% glutaraldehyde, 150 mM NaCl, 5 mM EGTA, 5 mM glucose, 5 mM $MgCl_2$ and 10 mM PIPES (pH 6.8) at room temperature for 10 min. Coverslips were washed twice with PBS and transferred to an imaging chamber for round coverglasses. Differential Interference Contrast (DIC) imaging was used to generate a map of the transfected cells. After confocal imaging, samples were fixed for EM in 2.5% glutaraldehyde, 2% formaldehyde, 2 mM $CaCl_2$ in 0.1 M cacodylate pH 7.4 for 15 min at room temperature followed by 45 min on ice. Coverslips were washed 4 times (5 min each) and post-fixed for 1 h at 4 °C in 0.25% $OsO_4$ and 0.25% potassium ferrocyanide in the same buffer. Samples were extensively washed with buffer, stained with 1% tannic acid for 1 h and 2% uranyl acetate in 50 mM acetate pH 5.5 for an additional hour. Samples were further dehydrated through a series of increasing concentrations of ethanol (twice with 50%, twice with 75% and 3 times with 100% anhydrous ethanol), and embedded in EMBed 812 epoxy resin (EMS). After resin polymerization for 60 h at 65 °C, the coverslips were removed with hydrofluoric acid, and cells previously imaged by light microscopy were identified by their position on the grid (for CLEM), cut out and remounted. Serial ultrathin sections (70–80 nm thick) were cut parallel to the plane of the coverslip and mounted on formvar/carbon coated slot (0.5 × 2 mm) EM grids. Sections were imaged in an FEI Tecnai 20 transmission electron microscope operated at 120 kV. Images were recorded on an AMT XR81 wide field CCD camera.

**Flow cytometry**. Neutral lipid content in WT and KO cells was measured by fluorescence-activated cell sorting (FACS) using BODIPY 493. Cells were washed, incubated with BODIPY 493 (stock solution in DMSO, used at 2 μg/ml in CM) for 25 min at 37 °C, washed, trypsinized, and resuspended in 1X PBS supplemented with 5 mM EDTA. FACS analysis was performed on an LSRFortessa flow cytometer (BD Biosciences). Viable cells were selected by gating according to an FSC-A vs. SSC-A plot (see Supplementary Fig. 2). Fluorescence intensities were analyzed using FlowJo software.

**FA pulse-chase**. HeLa cells were incubated in CM containing 1 μM RedC12 for 16 h at 37 °C. Cells were then washed three times with CM and further incubated for 1 h in CM. Cells were finally chased for the indicated times in CM or SM, fixed, permeabilized, and stained for the appropriate organelle markers (mitochondria with antibody to TOMM20, LDs with BODIPY 493) as described above.

**Mitochondrial respiration measurements**. Mitochondrial respiration was determined using a Seahorse Flux Analyzer XF96 (Agilent), according to the manufacturer's instructions. Briefly, 25,000 cells were seeded in each well of a Seahorse 96-well plate 2 days prior to the assay. One day before analysis, cells were starved in XF-based medium without phenol red (Agilent), pH 7.4, containing 0.5 mM glucose, 2 mM glutamine, 0.5 mM carnitine, and 1% FBS to deplete endogenous nutrient stores and prime cells for FA oxidation. On the day of the assay, cells were incubated for 1 h in 180 μl of XF-based medium without phenol red, pH 7.4 containing 111 mM NaCl, 4.7 mM KCl, 2.5 mM glucose, 0.5 mM carnitine, 5 mM HEPES. Etomoxir (Millipore Sigma, 4 μM) was added 15 min prior to measurement. For analysis of mitochondrial respiration driven by exogenous FAs, cells were incubated in CM containing 1 μM RedC12 or 1 μM OA for 16 h. Cells were then washed three times with CM, and further incubated for 1 h in CM prior to incubation with the substrate-limited medium. For measurements, cells were allowed to equilibrate for 1 h in a non-$CO_2$ incubator immediately before analysis of the metabolic flux. Oxygen consumption rates (OCR) were measured three times before sequential injection of 4 μM etomoxir (6 time points), 1 μM oligomycin A (Agilent, 3 time points), 1 μM FCCP (Agilent, 3 time points), and a mixture of 0.5 μM rotenone and antimycin A (Agilent, 3 time points). Three biological replicates were performed per cell line with eight technical replicates per sample. OCR values for each well were normalized according to the number of cells determined by the Cytation 1 reader (BioTek, Agilent). No significant changes in cell size were observed in the different cell lines and under the different conditions. Calculation of basal OCR before and after injection of etomoxir and non-mitochondrial OCR were made according to Agilent recommendations: basal OCR before etomoxir injection is the average of the technical replicates of the last time point (14.23 min), basal OCR after etomoxir injection is the average of the technical replicates of the last time point (37.47 min), non-mitochondrial OCR is the average of the technical replicates of the last time point (95.98 min). Basal OCR rates were calculated according to standard procedures, by subtracting the OCR insensitive to rotenone plus antimycin A.

**Tandem affinity purification and mass spectrometry (TAP-MS)**. HeLa cells stably expressing ATG9A-FTS were lysed in 50 mM Tris-HCl pH 7.4, 300 mM NaCl, 1 mM EDTA, and 1% Triton X-100 supplemented with protease inhibitor cocktail (Roche) for 30 min. Cell lysates were centrifuged at 17,000 x g for 10 min and clarified lysates were incubated overnight with Strep-Tactin resin (IBA) at 4 °C. Bound proteins were washed 3 times with 50 mM Tris-HCl pH 7.4, 300 mM NaCl, 1 mM EDTA, 0.1% Triton X-100, and eluted with 2.5 mM desthiobiotin (IBA). Proteins were further purified using FLAG M2 antibody-coated beads (Sigma). After incubation for 4 h at 4 °C, samples were washed 3 times with 50 mM Tris-HCl pH 7.4, 300 mM NaCl, 1 mM EDTA, 0.1% Triton X-100 and eluted twice with 500 μl of 500 μg/ml 3xFLAG peptide (Sigma). Proteins were precipitated with 10% trichloroacetic acid for 15 min at -20 °C, centrifuged for 30 min at 4 °C, washed twice with acetone, air-dried and analyzed by tandem mass spectrometry.

**Mass spectrometry**. TCA precipitated samples were resuspended in 50 μl 100 mM TEAB buffer, reduced with 5 mM TCEP for 1 h, alkylated with 10 mM NEM for 10 min, and digested with trypsin at 37 °C overnight. Peptides were desalted using Oasis HLB μElution plates (Waters). Digests of each sample were injected into an Ultimate 3000 RSLC nano HPLC system (Thermo Fisher Scientific). Peptides were separated on an ES802 column over a 63-min gradient with mobile phase B (98% acetonitrile, 1.9% $H_2O$, 0.1% formic acid) increased from 3% to 21%. LC-MS/MS data were acquired on an Orbitrap Lumos mass spectrometer (Thermo Fisher Scientific) in data-dependent acquisition mode. The MS1 scans were performed in orbitrap with a resolution of 120 K, a mass range of 200–1500 m/z, and an AGC target of $2 \times 10^5$. The quadrupole isolation was used with a window of 1.5 m/z. The MS/MS scans were triggered when the intensity of precursor ions with a charge state between 2 to 6 reached $1 \times 10^4$. The MS2 scans were conducted in ion trap. The HCD method was used with collision energy fixed at 30%. MS1 scan was performed every 3 s, and as many MS2 scans were acquired within the 3 s cycle.

The database search was performed using Proteome Discoverer 2.4 software. Mascot was used for sequence database search. Up to 2 missed cleavages were allowed for trypsin digestion. NEM on cysteines was set as static modification. Oxidation on methionine and acetyl on protein N-term were set as dynamic modifications. Mass tolerances for MS1 and MS2 scans were set to 5 ppm and 0.6 Da, respectively. Percolator was used for PSM validation. The search results were filtered by a false discovery rate of 1% at the protein level.

MS datasets were filtered for common contaminants against the Contaminant Repository for Affinity Purification Mass Spectrometry version 2.0 (CRAPome, www.crapome.org)[95] using two values: the "experiment appeared" ratio (ExpAP) (number of times the protein appeared in a CRAPome control experiment divided by the total number of CRAPome experiments analyzed) and the "average spectral count" (AveSC) (average number of peptides in the CRAPome control experiments where the protein appeared). The combination of ExpAP and AveSC provides an estimate of both the number of times the protein was identified in a CRAPome negative control and its abundance when it was identified.

**Strep-Tactin pulldown**. One 10-cm culture dish of cells per sample was transfected using Lipofectamine 2000. One day later, cells at 90% confluency were lysed in 20 mM HEPES, 200 mM NaCl, 1 mM DTT, and 1% DDM (n-dodecyl-β-D-maltopyranoside, Anagrade) supplemented with EDTA-free proteinase inhibitor cocktail (Roche) for 30 min. Cell lysates were cleared by centrifugation at 17,000 × g for 30 min at 4 °C and incubated with Strep-Tactin resin (IBA) overnight with agitation at 4 °C. Bound proteins were washed six times with 20 mM HEPES, 200 mM NaCl, 1 mM DTT, and 0.05% DDM. Beads were resuspended with 2x NuPage LDS sample buffer (Thermo Fischer Scientific) and incubated at 37 °C for 15 min. Beads were pelleted by centrifugation and eluates analyzed by SDS-PAGE and immunoblotting.

**GFP pulldown**. One 10-cm culture dish of cells per sample was transfected using Lipofectamine 2000. One day later, cells at 90% confluency were lysed in 20 mM HEPES, 200 mM NaCl, 1 mM DTT, and 1% DDM (n-dodecyl-β-D-maltopyranoside, Anagrade) supplemented with proteinase inhibitor cocktail EDTA-free (Roche) for 30 min at 4 °C, and incubated with GFP-Trap magnetic beads (Chromotek) for 3 h with agitation at 4 °C. Bound proteins were washed six times with 20 mM HEPES, 200 mM NaCl, 1 mM DTT, and 0.05% DDM. Beads were resuspended with 2x NuPage LDS sample buffer (Thermo Fischer Scientific) and incubated at 37 °C for 15 min. Beads were pelleted by centrifugation and eluates subsequently analyzed by SDS-PAGE and immunoblotting.

**C. elegans maintenance and CRISPR-Cas9 KO**. C. elegans strains (Supplementary Table 2) were maintained according to standard protocols. The AG608 atg9Δ strain was generated by CRISPR-Cas9 KO. The AG612 strain was generated by crossing AG444 (SEIP-1::mScarlet) males with DCR4521 hermaphrodites containing an endogenous knock-in atg-9(ola274[atg-9::gfp]) V by CRISPR-Cas9. F3 adults were screened for the presence of the SEIP-1::mScarlet and ATG-9::GFP.

The Bristol N2 strain was used as the WT for CRISPR-Cas9 genome editing. The atg-9-specific 20-nucleotide sequence for crRNA synthesis was selected using the guide RNA design tool from Integrated DNA Technologies (https://www.idtdna.com/site/order/designtool/index/CRISPR_CUSTOM) and synthesized as 4-nmol products by Dharmacon (https://horizondiscovery.com) along with tracRNA. Repair template design followed a standard protocol[96]. Detailed sequence information of the repair template and guide RNAs is listed below. Capitalized letters represent the ORF or exon sequence, lowercase letters indicate non-coding sequences, and bold letters indicate the nucleotides altered for the CRISPR design (Supplementary table 1).

Young gravid hermaphrodites (~20) were injected with mixed CRISPR-Cas9 reagents as described in the literature[97]. The atg-9(av244Δ) strain was generated by CRISPR-Cas9 mixes that contained two guide RNAs at flanking regions of the atg-9 coding region. Heterozygous atg-9Δ animals were first screened by PCR and then homozygosed in subsequent generations.

For brood size determinations, single mid-L4 hermaphrodites were picked onto 35 mm MYOB plates seeded with 10 μl of OP50 bacteria and allowed to lay eggs for 60 h. The brood sizes were determined at 60 h. The percentage of embryonic viability was calculated as the number of hatched larvae divided by the total number of hatched and unhatched animals multiplied by 100.

**C. elegans staining and imaging**. For staining of LDs in C. elegans, BODIPY 493 was dissolved in DMSO to 1 mg/ml. BODIPY 493 stock was diluted in M9 buffer (http://www.wormbook.org/chapters/www_strainmaintain/strainmaintain.html) to 6.7 μg/ml BODIPY 493 (final concentration of DMSO was 0.8%). Ten to 15 day 1 adult hermaphrodites (24 h post mid-L4) were washed in M9 three times and incubated in 6.7 μg/ml BODIPY 493 for 20 min in the dark, and washed again in M9 three times. All washes and incubations were performed in a concavity slide (Thermal Fisher Scientific, S99369). The stained hermaphrodites were anesthetized with freshly made 0.01% levamisole/M9 buffer for 15 min. The anesthetized animals were then transferred to 7% agarose pads for imaging. Image acquisition was conducted using a Nikon 60 × 1.2 NA water objective with a 0.3-μm z-step size. For imaging ATG-9::GFP expression, DCR4521 animals were immobilized as above, and DIC and GFP image acquisition were conducted using a Nikon 60 × 1.2 NA water objective with 1-μm z-step size; 15-20 planes were captured.

Live imaging was performed on a spinning-disk confocal system that includes a Nikon 60 × 1.2 NA water objective, a Photometrics Prime 95B EMCCD camera and a Yokogawa CSU-X1 confocal scanner unit. Images were acquired and analyzed by Nikon's NIS imaging software. LD diameters and numbers were quantified by Fiji/ImageJ Formats plugin[93,98]. The number of LDs was quantified in a unit volume (L 18 μm x W 18 μm x H 10 μm), and three independent units were measured in each animal.

**Ethical statement**. Our study protocol for C. elegans followed all NIH ethical regulations. All research conducted in this study has been approved by the Intramural Research Program of NIDDK.

**Quantification, reproducibility, and statistics**. All experimental data shown in this report, including micrographs, are representative from at least two independent experiments with similar results. No calculations were performed to

predetermine sample size. The sample size was determined based on similar experiments previously conducted by our group and others in similar fields. In fluorescence microscopy experiments, we quantified at least 20 cells randomly selected per condition from at least two independent replicates. Transfected cells expressing high level of proteins were excluded from analyses to avoid over-expression artifacts (pre-established criterion). Blinding was not done, as all fluorescence microscopy experiments were done by the same investigator. Quantification of immunoblotting and immunoprecipitation results was done from non-saturated images of at least three independent replicates. Seahorse experiments were also done in triplicate.

Analysis and plotting of data were performed using GraphPad Prism version 8.3.1 (GraphPad Software) and expressed as mean ± SD. The statistical significance of differences among multiple samples was assessed by one-way analysis of variance (ANOVA) and a Tukey post-hoc test. The unpaired Student's $t$-test was used for comparison of two groups. Statistical significance of *C. elegans* measurements was determined using an unpaired two-tailed $t$-test and both the Shapiro-Wilk and Kolmogorov-Smirnov Normality tests. The results were considered significantly different when $p < 0.05$ (*), $p < 0.01$ (**), $p < 0.001$ (***), or $p < 0.0001$ (****).

**Reporting summary**. Further information on research design is available in the Nature Research Reporting Summary linked to this article.

## Data availability

The mass spectrometry raw data generated in this study have been deposited in the MassIVE database and can be accessed as PXD029021. The processed mass spectrometry data are available as Supplementary Data 1. The microscopy data that support our findings are available on request from the corresponding author. Source data for Figs. 1–9 and Supplementary Figs. 1–8 are provided with the paper.

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

## Acknowledgements

We thank Noboru Mizushima, Gia Voeltz, and Feng Zhang for gifts of reagents, the *Caenorhabditis* Genetics Center, which is funded by the NIH Office of Research Infrastructure Programs (P40OD010440), and other members of the Bonifacino lab for helpful discussions. This work was supported by the Intramural Programs of NICHD (ZIA HD001607) (to J.S.B.) and NIDDK (to A.G.).

## Author contributions

J.S.B., E.M. and C.M.G. conceived the project and designed experiments. J.S.B. and E.M. wrote the manuscript. E.M. conducted experiments, analyzed data, and prepared figures. C.D.W. helped design the CLEM experiments and generated EM reconstructions. M.J. conducted electron microscopy. N.M. helped with Seahorse assays. X.B. and A.G. conducted the *C. elegans*, studies. Y.L. performed the mass spectrometry analysis of tandem affinity purification samples.

## Competing interests

The authors declare no competing interests.
