## [Peer Review File · Nature Communications]

The autophagy protein ATG9A enables lipid mobilization from lipid dropletsREVIEWER COMMENTS

Reviewer #1 (Remarks to the Author):

ATG9A is the only transmembrane protein in the autophagic machinery, and as such, has received much attention in the past years. Not only is the mechanism by which ATG9A functions unclear, but it has also been suggested that ATG9A, as well as other ATG proteins, may have roles unrelated to autophagy. In this paper, Mailler et al. suggests a new function of ATG9A, as a carrier of lipids from lipid droplets to the mitochondria as well as to the phagophore. Overall, this is an interesting and generally well performed study that implicating for the first time ATG9A in mitochondrial respiration. However, several issues require the authors attention:

- The authors use ATG7 KO cells as a control cell line, and to demonstrate that the observed phenotype is not the result of lipophagy. However, in figure 1b, a faint band can be detected for the ATG7 KO cells with an ATG7 antibody. Given that ATG7 is an enzyme, it is possible that low levels of it in the cell are sufficient for proper autophagy to take place. It is therefore important that the authors demonstrate the ATG7 has indeed been knocked out and that autophagy is impaired in this cell line.
- The controls in the colocalization analysis- rotating one of the channels by 90° and measuring Pearson's is somewhat indirect, additional complementary analysis is needed.
- In figure 2i, the authors use part of the images as a control because it seems like these parts show non-transfected cells. However, it is possible that these cells are transfected, but that the signal is below the detection limit. A control of cells transfected with a mock plasmid would serve this purpose.
- In figure 2, the authors demonstrate that there is an increase in the number of PLIN3 puncta in ATG9A KO cells, indicating more LD in those cells. However, WB analysis in the same figure shows no differences in the levels of PLIN3 in ATG9A KO cells compared to WT cells, and no explanation is given to explain this finding.
- In figure 4c, puncta are visible for RedC12 in SM with BODIPY493, but for RedC12 with TOM20 there are no RedC12 puncta- this requires explanation.
- There is no control sample for the experiment depicted in figure 6i.
- The authors describe an increase in LD size in animals, but not a change in LD number (which was observed in cell lines) and provide no explanation for the difference.
- In extended data figure 2- please provide zoom-in images as the colocalization is not clear.

Reviewer #2 (Remarks to the Author):

The authors demonstrate that ATG9A KO in HeLa cells results in an expansion of LD size and number that is independent of inhibition of autophagy (ATG7 KO) or lipophagy. Oleate loading and nutrient starvation conditions suggest that ATG9A regulates lipolysis of TAG on LD for mobilization. The defect in FA mobilization also impacts FA delivery to the mitochondria. ATG9A KO cells are also defective in FA oxidation, although the route for FA delivery to the mitochondria is unclear. Imaging studies reveal the ATG9A is associated with clusters of VTCs that are positioned between the ER, LDs and mitochondria. While these VTCs seem to be in close contact with these organelles (MCS??), there was minimal overlap with ATG2. ATG9A appears to localize and interact with TMEM41B on an ER subdomain suggesting a role in phagophore biogenesis. The model of how ATG9A is affecting lipid flux from LDs to other organelles places it in a dynamic VTC structure that apparently can contact other organelles and scramble phospholipids to facilitate transfer to target compartments.

While some of the data is intriguing, especially the localization of ATG9A to VTCs in proximity to LDs, Mito and ER, how it is specifically involved in lipid flow to these organelles is unclear. For instance, if ATG9A is involved in TAG hydrolysis and release of FA, these lipids would appear to have a couple of fates; either oxidation in mitochondria or conversion to phospholipids for incorporation into autophagosome. It is entirely unclear how ATG9A could coordinate this activity from the VTC and mediate lipid transfer to organelles if it has minimal interaction with ATG2 (or as yet unidentified lipid transfer proteins). This lack of mechanistic insight is exemplified by Figure 8E, which shows a very ambiguous role for ATG9A in the flux of FAs or phospholipids from LDs to other organelles.

The authors indicate the ATG9A 'likely' positivity regulates lipase activity on LDs. While this is partially supported by data in Figure 4, showing the lack of release of RedC12 from LDs under starvation conditions in ATG9A KO cells, this does not indicate why LDs accumulate under basal conditions. A more complete analysis of FA incorporation and release from LDs in control and KO cells might address this issue.

A plausible explanation for the LD phenotype is that ATG9A has an essential role in regulating the trans-bilayer distribution of phospholipids in the ER and other organelles. Its absence would disrupt PL distribution, potentially resulting in LDs with abnormal phospholipid composition that are resistant to lipolysis. Alternately, ATG9A KO leads to membrane stress and a preferred flux of FA into TAG versus PLs that is independent of any role it has in autophagy. Since ATG9A does not appear to directly contact these organelles and has minimal interaction with ATG2 and TMEM41B, this pleiotropic effect of KO on lipid homeostasis is a plausible explanation for the LD phenotype.

The title of the paper suggests that ATG9A is involved 'at sites' of phagophore formation but there is little evidence that it is physically positioned at an expanding phagophore. While they observe ATG9A in association with TMEM41B at the ER and in proximity to ER and mitochondria, there is no evidence from

CLEM or other IF imaging that ATG9A is adjacent to a phagophore. While this might be difficult to visualize, one would assume that ATG2A would be located at a phagophore given its role in lipid transport to this structure. However, there is very little overlap with ATG2 signals.

Minor point: Page 16, line 432-434, This reference (82) is from work in psd1 null yeast. In mammalian cells, it is apparent the PSD1 is essential for mitochondrial survival and PE synthesis in the ER cannot rescue.

Reviewer #3 (Remarks to the Author):

The paper entitled “The autophagy protein ATG9A enables lipid mobilization from lipid droplets at sites of phagophore formation” by Mailler et al. finds that loss of either ATG9A/B or ATG2A/B (proteins essential for autophagosome formation) in mammalian cell line leads to a dramatic expansion of lipid droplets (LD). The authors further characterised this phenotype and find that is not a result of general defects in autophagy of LD, hinting at a role for ATG9A and ATG2A/B in LD homeostasis and function. The very well written manuscript presents several lines of evidence supporting a role for ATG9A in fatty acid provision to the mitochondria via its lipid transport function. This direct model is very attractive and novel. Yet, many alternative and more indirect models could be drawn from the data. While excluding all alternative models is likely beyond feasible, it is crucial that no data goes against the proposed model, which is not the case here (see point 2, 3 5).

Major points

1. Throughout the manuscript the authors propose that ATG9A is required for membrane provision from LD to autophagosomes. Indeed, this is the title of the manuscript; however, it is not clear where this conclusion comes from, especially as two main figures are dedicated to the role of ATG9A in fatty acid provision to the mitochondria for β -oxidation. There is no data about provision of lipids from the LDs to the autophagosome.

2. The respiration data paint a picture that is incompatible with the authors model. Indeed, they observe that respiratory rate in WT cells can be increased by the addition of a fatty acid analog (RedC12), but not ATG9 or ATG2 mutant cells. The interpretation is that the analog is being fed to beta-oxidation to serve as substrate for respiration. Indeed, an inhibitor (Etomoxir) of the acylcarnitine system that shuttles fatty acids into mitochondria, decreases respiration rates of WT cells (RedC12-treated or not) but has no effect on ATG9 nor ATG2 KO cells, suggesting that Fatty acids shuttled by the acylcarnitine system fuel respiration in WT cells but fail to do so in mutant cells. However, subsequent FCCP treatment shows

much higher maximal uncoupled OCR in RedC12-treated WT cells, even though these cells have been previously treated by the acylcarnitine shuttle inhibitor. These data demonstrate that RedC12 affects OCR in conditions where fatty acids cannot serve as substrate for respiration, calling into question the interpretation of the experiment as a whole.

3. A related question is whether RedC12 can at all be used as substrate for beta-oxidation at all since it bears a large BODIPY molecule conjugated to the end of the fatty acyl chain? If not, then how does RedC12 affect oxygen consumption? Shouldn't the experiment be repeated with a physiological substrate of beta-oxidation like oleic acid?

4. The most spectacular and unexpected finding is that Atg2 or ATG9 mutants fail to import RedC12 into mitochondria upon amino acid starvation. The interpretation, along with the OCR experiments, is that RedC12 uses the acylcarnitine shuttle to get into mitochondria. Another possibility is that RedC12 has been incorporated into phospholipids and then transferred to mitochondria. The proper way to test it would be to purify mitochondria, and separate lipids by thin layer chromatography, to assess if the RedC12 is part of the phospholipid pool or comigrates with acylcarnitine. The simpler way of testing if RedC12 integrates mitochondria via the acylcarnitine shuttle, is to ask if Etomoxir treatment phenocopies atg2/9 KO in preventing RedC12 accumulation in mitochondria upon starvation.

5. If RedC12 can be utilised for β -oxidation, then what happens to the BODIPY dye at the end of the molecule after degradation of the acyl part of the molecule? Does it then localise to LD, as expected? LD are massively expanded in Atg9a KO cells, so is the BODIPY being soaked up there, hence leading to apparent lack of mitochondrial signal?

6. The low number of LD in the starved wild-type cells in figure 4B is confusing as other data in the manuscript show an increase in LD in this condition. Granted the timescales in figure 3 and figure 4 are different. Yet, if this is a reproducible phenomenon, it means that the LD increase upon starvation is, at some point reversed. How does that fit with the proposed model? What happens in the ATG2/9 KO cells? Doesn't the model imply that LD accumulation is, at least partly, a result of decreased transfer to mitochondria, and therefore that mitochondria transfer should be observed before LD accumulation (and then disappearance), and not 22h later?

7. P8 lines 191-196, the argument does not hold water. There is no reason to believe that ATG9 KO does not increase LD size/number by increasing the supply of FA. Moreover, the argument that cells have reached their maximum LD storage capacity because the effects of both treatments (Oleic acid and ATG9 KO) are not additive is tenuous at best. One could also argue that ATG9 is necessary for externally added oleic acid to reach LDs, and the effect are not additive because both affect the same process.

Responses to Reviewers

Reviewer #1:

ATG9A is the only transmembrane protein in the autophagic machinery, and as such, has received much attention in the past years. Not only is the mechanism by which ATG9A functions unclear, but it has also been suggested that ATG9A, as well as other ATG proteins, may have roles unrelated to autophagy. In this paper, Mailler et al. suggests a new function of ATG9A, as a carrier of lipids from lipid droplets to the mitochondria as well as to the phagophore. Overall, this is an interesting and generally well performed study that implicating for the first time ATG9A in mitochondrial respiration.

We thank the reviewer for finding our study “interesting and generally well performed”.

However, several issues require the authors attention:

- *The authors use ATG7 KO cells as a control cell line, and to demonstrate that the observed phenotype is not the result of lipophagy. However, in figure 1b, a faint band can be detected for the ATG7 KO cells with an ATG7 antibody. Given that ATG7 is an enzyme, it is possible that low levels of it in the cell are sufficient for proper autophagy to take place. It is therefore important that the authors demonstrate the ATG7 has indeed been knocked out and that autophagy is impaired in this cell line.*

The very faint band detected in ATG7-KO cells in Fig. 1b is likely non-specific, as the ATG7-KO cells show no conversion of LC3B-I to LC3B-II, even upon treatment with bafilomycin A1. This evidence, now shown in a new Supplementary fig. 1c, demonstrates complete absence of ATG7 activity in the KO cells.

- *The controls in the colocalization analysis- rotating one of the channels by 90° and measuring Pearson's is somewhat indirect, additional complementary analysis is needed.*

Use of misregistered images is a well-established method to control for random co-localization in fluorescence microscopy experiments (e.g., see Estakhr *et al*¹ Fig. 3D; Wand *et al*² Fig. 4F; reviewed in ref. ³). Therefore, to further validate the non-randomness of our co-localization data, we now show in the new Supplementary fig. 6a-h, another control in which the red channel was rotated 90° *counter-clockwise* relative to the green channel. Quantification shows that this rotation decreases the Pearson's coefficient in all cases. Together with the original control involving 90° *clockwise* rotation of the red channel, this new control confirms the specificity of the co-localizations shown in Fig. 6a-h.

- *In figure 2i, the authors use part of the images as a control because it seems like these parts show non-transfected cells. However, it is possible that these cells are transfected, but that the signal is below the detection limit. A control of cells transfected with a mock plasmid would serve this purpose.*

To address this comment, we now show as an additional control in Fig. 2i that expression of GFP does not rescue the LD phenotype in ATG9A-KO cells.

- *In figure 2, the authors demonstrate that there is an increase in the number of PLIN3 puncta in ATG9A KO cells, indicating more LD in those cells. However, WB analysis in the same figure shows no differences in the levels of PLIN3 in ATG9A KO cells compared to WT cells, and no explanation is given to explain this finding.*

We now mention in the text that these observations imply dilution of PLIN3 at the LD surface. We also cite two publications reporting similar observations upon incubation of

cells with oleic acid (OA) (bioRxiv preprint doi: <https://doi.org/10.1101/812990>, Fig. 2A and Fig. 6B; Hall et al⁴, Fig. 2C).

- In figure 4c, puncta are visible for RedC12 in SM with BODIPY493, but for RedC12 with TOM20 there are no RedC12 puncta- this requires explanation.

These differences are due to variations in the amount of RedC2 remaining in LDs in different cells or fields. We have now replaced in Fig. 4b images for the RedC12-BODIPY 493 pair by another set of images that is more comparable to the RedC12-TOMM20 pair. In any event, what matters the most is the quantification of co-localization shown in Fig. 4c, which results from quantification of much larger number of cells than those shown in the images.

- There is no control sample for the experiment depicted in figure 6i.

We now show in Fig. 6i (top row), a control experiment showing that overexpression of GFP does not redistribute ATG9A and TMEM41B-FTS to cell vertices.

- The authors describe an increase in LD size in animals, but not a change in LD number (which was observed in cell lines) and provide no explanation for the difference.

We now mention in the Discussion that ATG9 KO results in expansion of the LD pool, which, in different cell lines and organisms, is manifested as increases in LD number, LD size, or both, for reasons that are not understood at this time.

- In extended data figure 2- please provide zoom-in images as the colocalization is not clear.

We now include a fourth column with zoomed-in images of the boxed areas in Supplementary fig. 5 (former extended data figure 2).

Reviewer #2:

The authors demonstrate that ATG9A KO in HeLa cells results in an expansion of LD size and number that is independent of inhibition of autophagy (ATG7 KO) or lipophagy. Oleate loading and nutrient starvation conditions suggest that ATG9A regulates lipolysis of TAG on LD for mobilization. The defect in FA mobilization also impacts FA delivery to the mitochondria. ATG9A KO cells are also defective in FA oxidation, although the route for FA delivery to the mitochondria is unclear. Imaging studies reveal the ATG9A is associated with clusters of VTCs that are positions between the ER, LDs and mitochondria. While these VTCs seem to be in close contact with these organelles (MCS??), there was minimal overlap with ATG2. ATG9A appears to localize and interact with TMEM41B on an ER subdomain suggesting a role in phagophore biogenesis. The model of how ATG9A is affecting lipid flux from LDs to other organelles places it in a dynamic VTC structure that apparently can contact other organelles and scramble phospholipids to facilitate transfer to target compartments.

While some of the data is intriguing, especially the localization of ATG9A to VTCs in proximity to LDs, Mito and ER, how it is specifically involved in lipid flow to these organelles is unclear. For instance, If ATG9A involved in TAG hydrolysis and release of FA, these lipids would appear to have a couple of fates; either oxidation in mitochondria or conversion to phospholipids for incorporation into autophagosome. It is entirely unclear how ATG9A could coordinate this activity from the VTC and mediate lipid transfer to organelles if it has minimal interaction with ATG2 (or as yet unidentified lipid transfer proteins).

We agree that it is unclear how ATG9A promotes lipid transfer to mitochondria, and we now make this more explicit in the Discussion. The fact remains, however, that ATG9A is absolutely required for this transport, a surprising finding that has not been previously reported.

We would not characterize the interaction of ATG9A with ATG2 as “minimal”. Several groups, including ours, have previously shown that ATG9 physically interacts with ATG2 (Gomez-Sanchez *et al*⁵, Fig. 1A; Tang *et al*⁶, Figs. 6A and S7B; Ghanbarpour *et al*⁷, Fig. 2A; Guardia *et al*⁸, Fig. 5G), an observation that we again show in the new Supplementary fig. 6i,k panels of the present study. This interaction is consistent with the recent understanding that ATG2 channels lipids for ATG9 to scramble them at the VTC-phagophore interface⁹.

Concerning the limited (though significant) co-localization between ATG9A and ATG2 found in our study (Fig. 6a, new Supplementary fig. 5c, new Supplementary fig. 6a,b, new Supplementary fig. 7, new Supplementary fig. 6), we think it reflects the targeting of a large fraction of overexpressed ATG2A to LDs in addition to pre-autophagosomal structures. This LD targeting of ATG2A was first reported by the Mizushima lab^{10,11}, and confirmed in our study. The association of ATG2A with LDs is consistent with the affinity of ATG2A for lipids, although at present it is unknown if it is physiologically relevant or an artifact of overexpression. Unfortunately, to date, nobody has been able to stain endogenous ATG2A. Nevertheless, we wish to emphasize that the partial co-localization of ATG9A with ATG2A shown in our study is significant, in comparison to the previous and new random co-localization controls shown in Fig. 6a,b and new Supplementary fig. 6a,b. This is consistent with the physical interaction of ATG2A and ATG9A occurring at specific foci.

In our study, we also provide evidence for an ATG2-dependent interaction of ATG9A with the ER scramblase TMEM41B. This evidence includes IF co-localization, TAP-MS and co-immunoprecipitation. The co-localization of ATG9A-TMEM41B by IF microscopy, shown here for the first time, is striking, and likely pinpoints the sites where lipids are transferred from the ER to the ATG9A VTCs. Since the original submission of our manuscript, evidence has accrued that TMEM41B is a lipid scramblase^{7,12,13}, consistent with the physical interaction and spatial coincidence of ATG9A and TMEM41B. The new version of our manuscript has been updated to mention these recent findings on TMEM41B and their connection to our study.

This lack of mechanistic insight is exemplified by Figure 8E, which shows a very ambiguous role for ATG9A in the flux of FAs or phospholipids from LDs to other organelles.

We agree with the reviewer that the scheme shown in Fig. 8e (now Fig. 8f) is short on mechanistic details. However, our intention is to provide a graphical summary of our findings, including the involvement of ATG9, ATG2 and TMEM41B in lipid mobilization from LDs for transport to phagophores and mitochondria, without getting into finer details that we do not know at this time.

The authors indicate the ATG9A ‘likely’ positivity regulates lipase activity on LDs. While this is partially supported by data in Figure 4, showing the lack of release of RedC12 from LDs under starvation conditions in ATG9A KO cells, this does not indicate why LDs accumulate under basal conditions. A more complete analysis of FA incorporation and release from LDs in control and KO cells might address this issue.

We reread our original manuscript and nowhere did we find mention that “ATG9A ‘likely’ positivity regulates lipase activity on LDs.” Rather, we said that “ATG9A and ATG2A/B mediate LD-to-mitochondria FA transport independently of lipophagy, likely involving hydrolysis of LD TAGs by cytoplasmic lipases”. It’s a subtle but important distinction, in that ATG9A does not regulate lipolysis *per se* but depends on it for its role in lipid mobilization from LDs.

Nevertheless, the suggestion to examine lipid release from LDs in control and ATG9A-KO cells under basal conditions is well taken. We attempted a RedC12 chase experiment in WT and ATG9A-KO cells in complete medium. However, photobleaching of RedC12 over the course of the experiment preempted accurate quantification of differences in RedC12 decreases. We then turned to an alternative approach in which cells were pre-loaded with oleic acid and then incubated for different times in the absence of oleic acid in complete (i.e., not starvation) medium, followed by staining of neutral lipids with BODIPY 493. The results, shown in the new Supplementary fig. 3, demonstrate that upon removal of oleic acid, BODIPY 493 staining decreases over a 10 h-period in WT but not ATG9A-KO cells, revealing a defect in the mobilization of accumulated lipids from LDs in the KO cells. These results are described in connection with the oleic acid experiments that were already shown in Fig. 3a-c.

A plausible explanation for the LD phenotype is that ATG9A has an essential role in regulating the trans-bilayer distribution of phospholipids in the ER and other organelles. Its absence would disrupt PL distribution, potentially resulting in LDs with abnormal phospholipid composition that are resistant to lipolysis.

We thank the reviewer for suggesting this alternative explanation. We have extensively revised the Discussion to include this possible explanation.

Alternately, ATG9A KO leads to membrane stress and a preferred flux of FA into TAG versus PLs that is independent of any role it has in autophagy. Since ATG9A does not appear to directly contact these organelles and has minimal interaction with ATG2 and TMEM41B, this pleiotropic effects of KO on lipid homeostasis is a plausible explanation for the LD phenotype.

At the reviewer’s suggestion, we are also mentioning this alternative mechanism in the Discussion. With regard to the last sentence in this comment, we again wish to emphasize that we do not think that the interactions of ATG9A with ATG2 and TMEM41B are “minimal”. Properly controlled morphological and biochemical analyses demonstrate limited but significant interactions.

The title of the paper suggests that ATG9A is involved ‘at sites’ of phagophore formation but there is little evidence that it is physically positioned at a expanding phagophore. While they observe ATG9A in association with TMEM41B at the ER and in proximity to ER and mitochondria, there is no evidence from CLEM or other IF imaging that ATG9A is adjacent to a phagophore. While this might be difficult to visualize, one would assume that ATG2A would be located at a phagophore given its role in lipid transport to this structure. However, there is very little overlap with ATG2 signals.

We thank the reviewer for bringing up this issue. In response to this comment, we have deleted from the title “at sites of phagophore formation”, resulting in a more concise title. Nevertheless, in the revised version of our manuscript, we include an additional Supplementary fig. 7 showing two more CLEM images of ATG9A VTCs adjacent to phagophores/autophagosomes and to LDs containing associated ATG2A. Some of these phagophores/autophagosomes have a multilamellar structure, which is similar to that of other phagophores/autophagosomes reported in refs.¹⁴⁻¹⁶. To our knowledge, our

findings are the first to show ATG2A in the vicinity of phagophore-formation sites in mammalian cells. Our findings also support the notion that, in mammalian cells, ATG9A resides in VTCs that are adjacent to phagophores and is not integrated into the phagophore membrane. Because of its low throughput, CLEM is not a good method to find membrane-contact sites; this will require other methodologies that are currently beyond our possibilities.

Minor point: Page 16, line 432-434, This reference (82) is from work in psd1 null yeast. In mammalian cells, it is apparent the PSD1 is essential for mitochondrial survival and PE synthesis in the ER cannot rescue.

The reviewer correctly points out that the cited Psd1 study is in yeast and that the requirement for ER PE synthesis may be different in mammals. We have therefore deleted the paragraph dealing with mitochondrial PE, and folded this argument into another paragraph with a more general discussion of indirect effects that could explain our results.

Reviewer #3:

The paper entitled “The autophagy protein ATG9A enables lipid mobilization from lipid droplets at sites of phagophore formation” by Mailler et al. finds that loss of either ATG9A/B or ATG2A/B (proteins essential for autophagosome formation) in mammalian cell line leads to a dramatic expansion of lipid droplets (LD). The authors further characterised this phenotype and find that is not a result of general defects in autophagy of LD, hinting at a role for ATG9A and ATG2A/B in LD homeostasis and function. The very well written manuscript presents several lines of evidence supporting a role for ATG9A in fatty acid provision to the mitochondria via its lipid transport function. This direct model is very attractive and novel.

We thank this reviewer’s comment that our manuscript is “very well written” and that the “direct model is very attractive and novel.”

Yet, many alternative and more indirect models could be drawn from the data. While excluding all alternative models is likely beyond feasible, it is crucial that no data goes against the proposed model, which is not the case here (see point 2, 3 5).

We now discuss alternative models in the Discussion, as suggested by this reviewer and the other reviewer.

Major points

1. Throughout the manuscript the authors propose that ATG9A is required for membrane provision from LD to autophagosomes. Indeed, this is the title of the manuscript; however, it is not clear where this conclusion comes from, especially as two main figures are dedicated to the role of ATG9A in fatty acid provision to the mitochondria for β -oxidation. There is no data about provision of lipids from the LDs to the autophagosome.

The conclusion that ATG9A is required for lipid transport for LDs to autophagosomes is based on previous work by many groups. This work includes the demonstration that LDs are an important source of lipids for expansion of the autophagosome membrane^{17,18}. More recently, the TMEM41B-ATG2-ATG9 module was implicated in the delivery of lipids for expansion of the autophagosome membrane^{7,11,12,19-25}. Thus, although we do not directly show transport of lipids from LDs to autophagosomes (this would be difficult because of the transient nature of autophagosomes, as opposed to

mitochondria), it is the natural conclusion from many other studies. The unexpected and highly significant observation in our study is that ATG9A is required for non-autophagic lipid mobilization from LDs to mitochondria, demonstrating the importance of ATG9A in lipid transport processes other than those involved in autophagy..

2. *The respiration data paint a picture that is incompatible with the authors model. Indeed, they observe that respiratory rate in WT cells can be increased by the addition of a fatty acid analog (RedC12), but not ATG9 or ATG2 mutant cells. The interpretation is that the analog is being fed to beta-oxidation to serve as substrate for respiration. Indeed, an inhibitor (Etomoxir) of the acylcarnitine system that shuttles fatty acids into mitochondria, decreases respiration rates of WT cells (RedC12-treated or not) but has no effect on ATG9 nor ATG2 KO cells, suggesting that Fatty acids shuttled by the acylcarnitine system fuel respiration in WT cells but fail to do so in mutant cells. However, subsequent FCCP treatment shows much higher maximal uncoupled OCR in RedC12-treated WT cells, even though these cells have been previously treated by the acylcarnitine shuttle inhibitor. These data demonstrate that RedC12 affects OCR in conditions where fatty acids cannot serve as substrate for respiration, calling into question the interpretation of the experiment as a whole.*

We respectfully disagree that the respiration data are incompatible with our model. The observations that ATG9A-KO and ATG2A-B-KO cells have reduced basal OCR, and that etomoxir causes a similar reduction in WT basal OCR, are all consistent with ATG9A-KO and ATG2A-B-KO having lowered β -oxidation due to impairment of FA import into mitochondria (Fig. 5a,e). Moreover, addition of RedC12 causes a further etomoxir-sensitive increase in basal OCR in WT but not ATG9A-KO and ATG2A-B-KO cells. All of these results are fully consistent with our model.

The reviewer correctly points out that the maximal OCR revealed by treatment with FCCP is higher in WT than in ATG9A-KO and ATG2A-B-KO cells despite the continued presence of etomoxir in the medium (Fig. 5b-d,e). However, this finding can be explained by the experimental conditions of the assays. Etomoxir was injected 15 minutes prior to the start of the OCR recording. Maximal OCR attained by addition of FCCP mimics a physiological energy demand, which stimulates the respiratory chain to operate at maximum capacity. FCCP treatment causes rapid oxidation of pre-existing substrates (i.e., lipids, sugars, amino acids and reducing equivalents in the form of NADH and FADH₂). The larger burst in OCR in WT cells loaded with RedC12 upon addition of FCCP could result from a larger pool of FAs and its derivatives NADH and FADH₂ in mitochondria prior to the addition of etomoxir. This explanation is consistent with data in Fig. 2b of a previous report by Weisel *et al*²⁶. This effect does not negate the finding that etomoxir abrogates the differences in basal OCR between WT and KO cells, in line with the impaired import of FAs into mitochondria in the KO cells.

3. *A related question is whether RedC12 can at all be used as substrate for beta-oxidation at all since it bears a large BODIPY molecule conjugated to the end of the fatty acyl chain? If not, then how does RedC12 affect oxygen consumption? Shouldn't the experiment be repeated with a physiological substrate of beta-oxidation like oleic acid?*

A previous study showed that RedC12 is a suitable substrate for mitochondrial β -oxidation²⁷. In that study, thin-layer chromatography analyses showed that RedC12 is esterified, and that starvation causes a decrease in the levels of esterified RedC12 concomitant with the appearance of RedC12 breakdown products, all inhibitable by etomoxir²⁷. Nevertheless, to further address this reviewer's comment, we conducted Seahorse experiments to compare RedC12 with oleic acid as substrates for β -oxidation and found that both are equally effective at raising the basal OCR (shown in the new Supplementary fig. 4c). These findings further support the use of RedC12 in the

Seahorse experiments. Moreover, RedC12 was chosen as a substrate in these experiments for consistency with the RedC12 transport experiments shown in Fig. 4.

4. *The most spectacular and unexpected finding is that Atg2 or ATG9 mutants fail to import RedC12 into mitochondria upon amino acid starvation. The interpretation, along with the OCR experiments, is that RedC12 uses the acylcarnitine shuttle to get into mitochondria. Another possibility is that RedC12 has been incorporated into phospholipids and then transferred to mitochondria. The proper way to test it would be to purify mitochondria, and separate lipids by thin layer chromatography, to assess if the RedC12 is part of the phospholipid pool or comigrates with acylcarnitine. The simpler way of testing if RedC12 integrates mitochondria via the acylcarnitine shuttle, is to ask if Etomoxir treatment phenocopies atg2/9 KO in preventing RedC12 accumulation in mitochondria upon starvation.*

Biochemical analyses of RedC12 incorporation into phospholipids or acylcarnitine are beyond our possibilities at this time. However, we did perform the alternative experiment suggested by the reviewer. The experiment consisted of incubating WT cells in complete medium or starvation medium in the presence of etomoxir, and quantifying the localization of RedC12 relative to BODIPY 493 and TOMM20. We observed that etomoxir blocked the transfer of RedC12 from LDs to mitochondria under starvation conditions (new Supplementary fig. 4a,b), thus phenocopying the effect of ATG9A KO or ATG2A-B KO.

5. *If RedC12 can be utilised for β -oxidation, then what happens to the BODIPY dye at the end of the molecule after degradation of the acyl part of the molecule? Does it then localise to LD, as expected? LD are massively expanded in Atg9a KO cells, so is the BODIPY being soaked up there, hence leading to apparent lack of mitochondrial signal?*

We are not aware of any studies following the long-term metabolic fate of the BODIPY dye after β -oxidation of the RedC12 FA chain in mitochondria. We consulted the manufacturer, Thermo, and they said “The BODIPY dye will remain in any hydrophobic environment even without the acyl part of the molecule.” We agree with the reviewer that BODIPY 558/568 could partition back into LDs or become diluted in other membranes. Its concentration, however, would be too low for visualization (as opposed to the addition of excess, 2 μ g/ml BODIPY 493 used for LD labeling).

6. *The low number of LD in the starved wild-type cells in figure 4B is confusing as other data in the manuscript show an increase in LD in this condition. Granted the timescales in figure 3 and figure 4 are different. Yet, if this is a reproducible phenomenon, it means that the LD increase upon starvation is, at some point reversed. How does that fit with the proposed model? What happens in the ATG2/9 KO cells? Doesn't the model imply that LD accumulation is, at least partly, a result of decreased transfer to mitochondria, and therefore that mitochondria transfer should be observed before LD accumulation (and then disappearance), and not 22h later?*

This is a reproducible phenomenon. Starvation is known to cause an increase in the abundance of LDs in mouse embryonic fibroblasts^{27,28}. Fig. 3 of our study shows that this is also the case in the HeLa cells used in our study. However, we do see a decrease in the number and size of LDs upon long-term starvation in HeLa cells, as shown in Fig. 4. This is in agreement with other studies in HeLa and other cell lines, which showed a “bell-shaped” curve for the number of LDs as a function of time in starved cells²⁹. So, the differences in LD number/size in Figs. 3 and 4 of our paper are explained by the different times of starvation (0-3 h vs. 24 h, respectively). In ATG9A-KO or ATG2A-B-KO cells, the number/size of LD remains high (Fig. 4d) because of decreased lipid transfer to mitochondria (and likely to autophagosomes). The protocol that we used in

Fig. 4 is that previously established by others²⁷. A long starvation period is needed to observe accumulation of RedC12 in mitochondria.

7. P8 lines 191-196, the argument does not hold water. There is no reason to believe that ATG9 KO does not increase LD size/number by increasing the supply of FA.

We agree with the reviewer that the cited paragraph overinterpreted the data. We have therefore deleted most speculation and simply stated the conclusions from the experiments in a much shorter paragraph.

Moreover, the argument that cells have reached their maximum LD storage capacity because the effects of both treatments (Oleic acid and ATG9 KO) are not additive is tenuous at best. One could also argue that ATG9 is necessary for externally added oleic acid to reach LDs, and the effect are not additive because both affect the same process.

As described above, we have shortened this paragraph to remove unnecessary speculation. However, we have kept the statement about the treatments being largely non-additive because this is a direct observation from the experiments, irrespective of the mechanisms involved.

References

- 1 Estakhr, J., Abazari, D., Frisby, K., McIntosh, J. M. & Nashmi, R. Differential Control of Dopaminergic Excitability and Locomotion by Cholinergic Inputs in Mouse Substantia Nigra. *Curr Biol* **27**, 1900-1914 e1904, doi:10.1016/j.cub.2017.05.084 (2017).
- 2 Wang, J. et al. High Temporal Resolution Reveals Simultaneous Plasma Membrane Recruitment of TPLATE Complex Subunits. *Plant Physiol* **183**, 986-997, doi:10.1104/pp.20.00178 (2020).
- 3 Dunn, K. W., Kamocka, M. M. & McDonald, J. H. A practical guide to evaluating colocalization in biological microscopy. *Am J Physiol Cell Physiol* **300**, C723-742, doi:10.1152/ajpcell.00462.2010 (2011).
- 4 Hall, A. M. et al. Dynamic and differential regulation of proteins that coat lipid droplets in fatty liver dystrophic mice. *J Lipid Res* **51**, 554-563, doi:10.1194/jlr.M000976 (2010).
- 5 Gomez-Sanchez, R. et al. Atg9 establishes Atg2-dependent contact sites between the endoplasmic reticulum and phagophores. *J Cell Biol* **217**, 2743-2763, doi:10.1083/jcb.201710116 (2018).
- 6 Tang, Z. et al. TOM40 Targets Atg2 to Mitochondria-Associated ER Membranes for Phagophore Expansion. *Cell Rep* **28**, 1744-1757 e1745, doi:10.1016/j.celrep.2019.07.036 (2019).
- 7 Ghanbarpour, A., Valverde, D. P., Melia, T. J. & Reinisch, K. M. A model for a partnership of lipid transfer proteins and scramblases in membrane expansion and organelle biogenesis. *Proc Natl Acad Sci U S A* **118**, doi:10.1073/pnas.2101562118 (2021).
- 8 Guardia, C. M. et al. Structure of Human ATG9A, the Only Transmembrane Protein of the Core Autophagy Machinery. *Cell Rep* **31**, 107837, doi:10.1016/j.celrep.2020.107837 (2020).
- 9 Chang, C., Jensen, L. E. & Hurley, J. H. Autophagosome biogenesis comes out of the black box. *Nat Cell Biol* **23**, 450-456, doi:10.1038/s41556-021-00669-y (2021).
- 10 Tamura, N. et al. Differential requirement for ATG2A domains for localization to autophagic membranes and lipid droplets. *FEBS Lett* **591**, 3819-3830, doi:10.1002/1873-3468.12901 (2017).
- 11 Velikkakath, A. K., Nishimura, T., Oita, E., Ishihara, N. & Mizushima, N. Mammalian Atg2 proteins are essential for autophagosome formation and important for regulation

- of size and distribution of lipid droplets. *Mol Biol Cell* **23**, 896-909, doi:10.1091/mbc.E11-09-0785 (2012).
- 12 Li, Y. E. et al. TMEM41B and VMP1 are scramblases and regulate the distribution of
cholesterol and phosphatidylserine. *J Cell Biol* **220**, doi:10.1083/jcb.202103105 (2021).
- 13 Huang, D. et al. TMEM41B acts as an ER scramblase required for lipoprotein biogenesis
and lipid homeostasis. *Cell Metab*, doi:10.1016/j.cmet.2021.05.006 (2021).
- 14 Uemura, T. et al. A cluster of thin tubular structures mediates transformation of the
endoplasmic reticulum to autophagic isolation membrane. *Mol Cell Biol* **34**, 1695-1706,
doi:10.1128/MCB.01327-13 (2014).
- 15 Yla-Anttila, P., Vihinen, H., Jokitalo, E. & Eskelinen, E. L. 3D tomography reveals
connections between the phagophore and endoplasmic reticulum. *Autophagy* **5**, 1180-
1185, doi:10.4161/auto.5.8.10274 (2009).
- 16 Hayashi-Nishino, M. et al. A subdomain of the endoplasmic reticulum forms a cradle for
autophagosome formation. *Nat Cell Biol* **11**, 1433-1437, doi:10.1038/ncb1991 (2009).
- 17 Dupont, N. et al. Neutral lipid stores and lipase PNPLA5 contribute to autophagosome
biogenesis. *Curr Biol* **24**, 609-620, doi:10.1016/j.cub.2014.02.008 (2014).
- 18 Shpilka, T. et al. Lipid droplets and their component triglycerides and steryl esters
regulate autophagosome biogenesis. *EMBO J* **34**, 2117-2131,
doi:10.15252/embj.201490315 (2015).
- 19 Maeda, S., Otomo, C. & Otomo, T. The autophagic membrane tether ATG2A transfers
lipids between membranes. *Elife* **8**, doi:10.7554/eLife.45777 (2019).
- 20 Maeda, S. et al. Structure, lipid scrambling activity and role in autophagosome formation
of ATG9A. *Nat Struct Mol Biol* **27**, 1194-1201, doi:10.1038/s41594-020-00520-2 (2020).
- 21 Osawa, T. et al. Atg2 mediates direct lipid transfer between membranes for
autophagosome formation. *Nat Struct Mol Biol* **26**, 281-288, doi:10.1038/s41594-019-0203-
4 (2019).
- 22 Valverde, D. P. et al. ATG2 transports lipids to promote autophagosome biogenesis. *J
Cell Biol* **218**, 1787-1798, doi:10.1083/jcb.201811139 (2019).
- 23 Morita, K. et al. Genome-wide CRISPR screen identifies TMEM41B as a gene required for
autophagosome formation. *J Cell Biol* **217**, 3817-3828, doi:10.1083/jcb.201804132 (2018).
- 24 Moretti, F. et al. TMEM41B is a novel regulator of autophagy and lipid mobilization.
EMBO Rep **19**, doi:10.15252/embr.201845889 (2018).
- 25 Shoemaker, C. J. et al. CRISPR screening using an expanded toolkit of autophagy
reporters identifies TMEM41B as a novel autophagy factor. *PLoS Biol* **17**, e2007044,
doi:10.1371/journal.pbio.2007044 (2019).
- 26 Weisel, F. J. et al. Germinal center B cells selectively oxidize fatty acids for energy while
conducting minimal glycolysis. *Nat Immunol* **21**, 331-342, doi:10.1038/s41590-020-0598-4
(2020).
- 27 Rambold, A. S., Cohen, S. & Lippincott-Schwartz, J. Fatty acid trafficking in starved cells:
regulation by lipid droplet lipolysis, autophagy, and mitochondrial fusion dynamics.
Dev Cell **32**, 678-692, doi:10.1016/j.devcel.2015.01.029 (2015).
- 28 Nguyen, T. B. et al. DGAT1-Dependent Lipid Droplet Biogenesis Protects Mitochondrial
Function during Starvation-Induced Autophagy. *Dev Cell* **42**, 9-21 e25,
doi:10.1016/j.devcel.2017.06.003 (2017).
- 29 Cabodevilla, A. G. et al. Cell survival during complete nutrient deprivation depends on
lipid droplet-fueled beta-oxidation of fatty acids. *J Biol Chem* **288**, 27777-27788,
doi:10.1074/jbc.M113.466656 (2013).

REVIEWERS' COMMENTS

Reviewer #1 (Remarks to the Author):

The authors successfully addressed the concerns raised in the first round of review and in its present form the manuscript meets Nat. Comm. scientific merit

Reviewer #2 (Remarks to the Author):

Middle page 12. Looks like ATG9 is in close proximity to a mature autophagosomes/lysosomes but not a developing phagophore? Granted, the observation of ATG9 adjacent to phagophores has been shown elsewhere but the authors should better define what exactly is shown in the Supplemental Figure 7.

Bottom of page 15. I fail to see how ATG9 could mediate FA transfer to mitochondria to the ER given the lack of supporting evidence presented in the study. FA once released by lipolysis would be converted to CoA esters and delivered to the different organelles by diffusion or FA binding proteins. The scramblases don't transfer FA or CoA esters and don't appear to be on the surface of the mitochondria or LDs, suggesting they are not directly involved (ie. 'mediate' transfer) seems at odds with know functions.

Generally, the discussion is very speculative due to the lack of any hard mechanistic evidence of how lack of ATG9 could be involved in LD accretion.

Reviewer #3 (Remarks to the Author):

The revisions to the manuscript by Mailler et al. successfully address most previous comments. In particular, the fact that etomoxir phenocopies ATG2/Atg9 deletion in blocking the RedC12 fluorescence transfer to the mitochondria is compelling. There are, however, still concerns with the Seahorse data, as presented.

The author's explanation that "The larger burst in OCR in WT cells loaded with RedC12 upon addition of FCCP could result from a larger pool of FAs and its derivatives NADH and FADH2 in mitochondria prior to

the addition of etomoxir" isn't convincing. If there was a large pool of substrate present in the mitochondria before Etomoxir addition, then this treatment would have no effect, since according to the autor's model, etomoxir precisely blocks the step that RedC12 is supposed to stimulate (i.e. supply of respiratory substrate). Etomoxir effect should only be felt after the presumed "larger pool of FAs [...] in mitochondria" has been depleted.

Looking closely at the data it appears that much more is happening in the metabolic status of the cells than portrayed by the simple model put forward.

As stated in the original review, RedC12 does not only stimulate the basal OCR but also the maximal OCR (upon FCCP), and to a similar extent. In fact, when looking closely, RedC12 also stimulates non-mitochondrial oxygen consumption (after antimycin+rotenone) both in WT and ATG7KO, and both to a similar extent as basal OCR. Taken together, it appears that more complex changes in cell metabolism are happening. It is, for instance, possible that RedC12 addition messes up the normalization of the experiment, something that could be the result of slightly altered cell size or mitochondrial content (if RedC12-treated WT cells were 30% larger, all OCR measurements would be 30% higher since they are normalized by cell number). Finally, proton leak in ATG9/ATG2 knockout cells is much higher than wild-type cells, as observed by the difference between oligomycin treatment and Rotenone + Antimycin-A treatment, while RedC12 treatment appear to all but abolish leak current in WT and ATG7KO cells.

While it is clear however is that, contrary to WT and ATG7Ko cells, Etomoxir has no detectable effect on basal OCR in ATG2/9 KO cells, which should be emphasized, it is equally important to emphasize that the respiration data indicate broader metabolic adaptations that cannot be solely explained by the model put forward.

Responses to Reviewers

We thank the reviewers for their additional comments. New changes to the text are indicated in red font in the attached manuscript file.

Reviewer #1

The authors successfully addressed the concerns raised in the first round of review and in its present form the manuscript meets Nat. Comm. scientific merit.

We are pleased that we were able to fully address this reviewer's comments.

Reviewer #2

The authors should better define what exactly is shown in the Supplemental Figure 7.

In the new version of the Supplementary figures, we now include a full description of Supplementary fig. 7 in the corresponding legend.

Bottom of page 15. I fail to see how ATG9 could mediate FA transfer to mitochondria to the ER given the lack of supporting evidence presented in the study. FA once released by lipolysis would be converted to CoA esters and delivered to the different organelles by diffusion or FA binding proteins. The scramblases don't transfer FA or CoA esters and don't appear to be on the surface of the mitochondria or LDs, suggesting they are not directly involved (ie. 'mediate' transfer) seems at odds with know functions.

We have revised the discussion of potential mechanisms by which ATG9A, ATG2A/B and TMEM41B promote FA transport from LDs to mitochondria, now giving more weight to indirect mechanisms. We also mention the conversion of FAs to FA-CoA, and further emphasize fact that there is no evidence that ATG9A, ATG2A/B and TMEM41B directly transport FAs or FA-CoA. To avoid misinterpretation of our findings, we have deleted the scheme in Fig. 8.

Generally, the discussion is very speculative due to the lack of any hard mechanistic evidence of how lack of ATG9 could be involved in LD accretion.

See response to previous point. We think that the revised Discussion presents a more balanced consideration of both direct and indirect mechanisms that could explain our findings.

Reviewer #3

The revisions to the manuscript by Mailler et al. successfully address most previous comments. In particular, the fact that etomoxir phenocopies ATG2/Atg9 deletion in blocking the RedC12 fluorescence transfer to the mitochondria is compelling.

We are pleased that we have been able to address most previous comments.

There are, however, still concerns with the Seahorse data, as presented. The author's explanation that "The larger burst in OCR in WT cells loaded with RedC12 upon addition of FCCP could result from a larger pool of FAs and its derivatives NADH and FADH2 in mitochondria

prior to the addition of etomoxir" isn't convincing. If there was a large pool of substrate present in the mitochondria before Etomoxir addition, then this treatment would have no effect, since according to the autor's model, etomoxir precisely blocks the step that RedC12 is supposed to stimulate (i.e. supply of respiratory substrate). Etomoxir effect should only be felt after the presumed "larger pool of FAs [...] in mitochondria" has been depleted.

This point is well taken. We agree with the Reviewer that our explanation is not satisfactory. Since this explanation was not included in the manuscript but was for the Reviewer only, we did not make any changes to the text.

Looking closely at the data it appears that much more is happening in the metabolic status of the cells than portrayed by the simple model put forward.

As stated in the original review, RedC12 does not only stimulate the basal OCR but also the maximal OCR (upon FCCP), and to a similar extent. In fact, when looking closely, RedC12 also stimulates non-mitochondrial oxygen consumption (after antimycin+rotenone) both in WT and ATG7KO, and both to a similar extent as basal OCR. Taken together, it appears that more complex changes in cell metabolism are happening. It is, for instance, possible that RedC12 addition messes up the normalization of the experiment, something that could be the result of slightly altered cell size or mitochondrial content (if RedC12-treated WT cells were 30% larger, all OCR measurements would be 30% higher since they are normalized by cell number). Finally, proton leak in ATG9/ATG2 knockout cells is much higher than wild-type cells, as observed by the difference between oligomycin treatment and Rotenone + Antimycin-A treatment, while RedC12 treatment appear to all but abolish leak current in WT and ATG7KO cells.

While it is clear however is that, contrary to WT and ATG7Ko cells, Etomoxir has no detectable effect on basal OCR in ATG2/9 KO cells, which should be emphasized, it is equally important to emphasize that the respiration data indicate broader metabolic adaptations that cannot be solely explained by the model put forward.

We agree with the Reviewer that the effects of excess FA addition and ATG9A / ATG2A-B KO are complex, and that more is happening than just an increase in basal OCR. We now mention in the corresponding section of the Results that excess FAs increase not only basal OCR but also maximal OCR and non-mitochondrial oxygen consumption, and that ATG9A or ATG2A-B KO increase proton leak. We end this sections with the statement: "ATG9A-KO and ATG2A-B-KO cause additional alterations in energy metabolism besides the inhibition of FA transport from LDs to mitochondria."